# Narrative thinking lingers in spontaneous thought

Buddhika Bellana [1,2] ✉, Abhijit Mahabal[3] & Christopher J. Honey [1] ✉

Some experiences linger in mind, spontaneously returning to our thoughts for minutes after their conclusion. Other experiences fall out of mind immediately. It remains unclear why. We hypothesize that an input is more likely to persist in our thoughts when it has been deeply processed: when we have extracted its situational meaning rather than its physical properties or low-level semantics. Here, participants read sequences of words with different levels of coherence (word-, sentence-, or narrative-level). We probe participants' spontaneous thoughts via free word association, before and after reading. By measuring lingering subjectively (via self-report) and objectively (via changes in free association content), we find that information lingers when it is coherent at the narrative level. Furthermore, and an individual's feeling of transportation into reading material predicts lingering better than the material's objective coherence. Thus, our thoughts in the present moment echo prior experiences that have been incorporated into deeper, narrative forms of thinking.

Human thought is history-dependent: how we think and what we think about at any moment is shaped by what came before[1,2]. A simple example of this phenomenon can be seen in semantic priming, in which our ability to identify a given word is heightened following the activation of related concepts[3,4]. For example, it is easier to recognize the word "butter" after being exposed to the word "bread". A more sophisticated picture is described by theories of drifting mental context. Here, an internal representation of context is continually updated as we recursively encode and retrieve the moments of our lives[5,6]. These models of mental context are powerful because they explain how one thought becomes part of a broader trajectory[7]. Moreover, models of mental context can be extended to include many dimensions of mental context, accounting for influences in the semantic[8], spatial[9] and emotional[10] domains.

Given that our mental context has a wide-reaching influence on memory[11,12], comprehension[13] and decision-making[14], it is natural to ask: which kinds of mental content influence the trajectory of our thoughts most strongly? For example, avid readers report that the experience of a novel does not end upon closing the book, but can linger in their minds for hours or days[15,16]. Similarly, the content from role-playing video games also has a propensity to linger in mind[17,18].

More generally, social information[19,20] and emotional experiences[21–23] tend to exert long-lasting influences on our mental context, in many cases intruding on our thoughts against our will[24,25]. Intuitively, it seems that these meaningful, real-world experiences affect our subsequent thoughts in a manner that goes beyond lexical or semantic priming. But why should these particular types of processing – narrative, social and emotional – linger in our thoughts?

We hypothesized that the stream of our spontaneous thoughts is especially affected by "deeper" and more elaborated forms of mental processing. Here, shallower levels of processing correspond to extracting the immediate physical features of a stimulus or arbitrary associations decoupled from our world-knowledge, while deeper levels of processing entail extracting and representing more abstract features that concern what an input implies[26]. Moreover, we propose that the deepest levels of processing are those in which we access an especially rich bed of existing associations to contextualize an input, such as when we evaluate self-relevance[27] or build mental models of naturalistic situations[28–30]. For example, we can attend to the words that make up a novel, in terms of what they look like or their individual semantic meanings – but only when they are read in their broader context, and word meanings are considered in relation to one another,

[1]Department of Psychological & Brain Sciences, Johns Hopkins University, Baltimore, MD, USA. [2]Department of Psychology, Glendon Campus, York University, Toronto, ON, Canada. [3]Pinterest, San Francisco, CA, USA. ✉e-mail: bbellana@yorku.ca; chris.honey@jhu.edu

can we appreciate the complex progression of events, characters, goals, actions and emotions that they imply.

Lingering should be elicited both by the properties of the stimulus and by our appraisal of it[24,31]. Stories contain high-level semantics and situational information, and so the process of comprehending a narrative is likely to require deep meaning-centered processing, leading to lingering. However, deep processing can also arise from endogenous interests and dispositions toward the world: we may struggle to interest ourselves in an episode of a popular TV show, while at the same time finding ourselves engrossed in the plight of an ant struggling to carry an outsized breadcrumb down from our armchair. We predict that the extent to which we engage deeply in our thinking about the ant, specifically considering it in terms of the broader situation or narrativizing its activities (e.g., where is it going and why is it so motivated?), will cause ant-related thoughts to linger after the ant leaves our immediate perception.

We began testing our hypothesis that deep processing exerts a lasting influence on our spontaneous thoughts by manipulating the depth with which participants read short stories (~2500 words). For these studies employing narrative stimuli, we defined deep processing as the extraction of situation-level meaning from a text. We manipulated depth by presenting participants with either an intact narrative or the same text with the sentence- or word-order randomly scrambled, given that scrambling limits the extent of situation-level meaning a reader can readily extract from the text[32]. Before and after reading, participants performed a free association task, in which they freely typed words for five minutes. We then used document classification[33]

and natural language processing tools[34] to quantify the extent to which story themes were expressed in each participant's free associations, before and after reading.

When participants read coherent narratives, the themes and details of the story lingered for several minutes in their subsequent free association chains, more so than in participants who read scrambled versions of the same text. This observation replicated and generalized across multiple stories. A follow-up experiment demonstrated that coherent stories lingered more when participants judge the emotional properties of the sentences rather than their spelling or font, and also that the lingering was mostly nonvolitional. Moreover, the lingering effect was again observed when participants narrativized a word list, but not when they judged the perceptual properties of each individual word (e.g., italic type). Overall, regardless of the objective coherence of what participants read, their subsequent experience of lingering was best predicted by the degree to which they felt transported by the material while reading.

A "depth of processing" framework can account for our results. According to this framework, engaging with a text's typography is shallower processing than engaging with its word-level semantics[26] and the extraction of situation-level meaning from a text is deeper processing than extracting word- or even sentence-level semantics[29]. When applied to our data, a clear pattern emerges: deeper processing predicts greater subsequent lingering. More generally, our data indicate that more elaborative styles of thinking, such as the construction of situation models while reading stories, produce an especially long-lasting mental context.

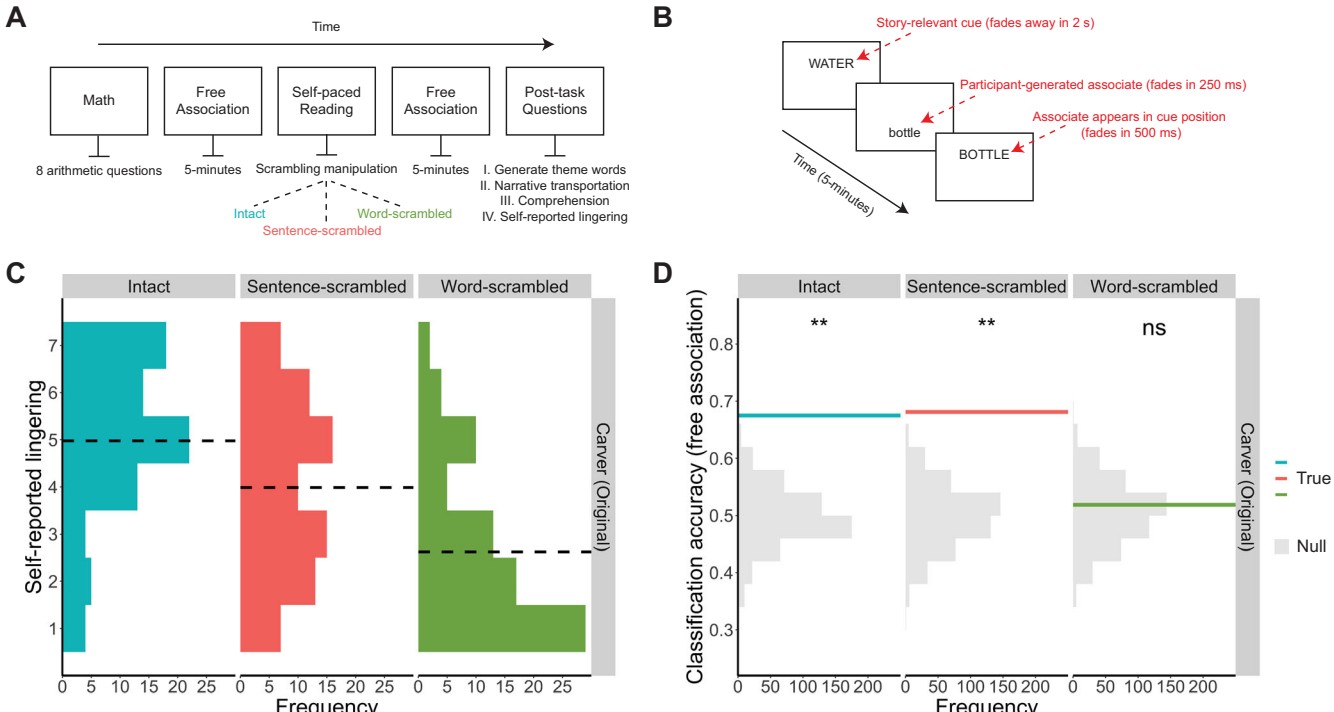

**Fig. 1 | Stories, not words, elicit a lasting influence on spontaneous thought. A** Schematic of the experimental paradigm. For details, see Methods and SI: Supplemental Methods. **B** Schematic of free association task. Participants freely typed words for 5-minutes, before and after reading the story. **C** Histograms of participants' responses to the question: "To what extent did the text linger in your mind after reading it?". Participants provided their rating on a 7-pt scale: 7 = very much, 1 = not at all. Black dashed line represents the mean rating per condition. $n = 80$ participants per condition. **D** Histograms of how accurately a document classifier could discriminate between pre- and post-story free association. Classifiers were trained within-condition ($n = 80$ participants), using a leave-one-participant-out cross-validation procedure with 500 bootstraps. Solid lines represent the mean classification accuracy. Null distributions were estimated by randomly shuffling the labels of the held-out participant's word chains (pre, post) and recalculating classification accuracy over 500 permutations. Likelihood of achieving mean classification from the null distribution was calculated using a one-sided permutation test [ns $p > 0.05$; *$p = <0.05$; ** $p = <0.01$; Note 1: all $p$s are uncorrected with respect to multiple comparisons; Note 2: minimum p-value estimate for this analysis is $p < 0.002$]. The exact accuracies and $p$ values for each condition are as follows: Intact [68% accuracy, $p < 0.002$], Sentence-scrambled [68% accuracy, $p < 0.002$]; Word-scrambled [52% accuracy, $p = 0.340$]. Source data for panels C and D are provided as a Source Data file.

## Results

Two-hundred and forty online participants read versions of the short story 'So Much Water So Close To Home' by Raymond Carver in one of three randomly assigned conditions (Intact: $n = 80$, Sentence-Scrambled: $n = 80$, Word-Scrambled: $n = 80$) (Fig. 1A). In each condition, participants were shown the exact same words, with the only difference being their order. Participants read the text at their own pace, one sentence at a time. They performed a 5-minute, unconstrained free association task before and after the reading (Fig. 1B). Following free association, participants described the core themes of the story and rated the extent they felt transported while reading the text. Next, participants completed a test of story comprehension and rated the extent to which the text lingered in their mind after reading.

### Scrambling limits deep processing

First, we confirmed that our scrambling procedure indeed limited the extent to which participants were able to extract situation-level meaning from the text. To this end, we examined a measure of narrative transportation, or participants' self-reported sense of being transported into the "world" of the story while reading.

Transportation was measured using a 13-item modified version of the Narrative Transportation scale[35]. Transportation requires participants to attend to deeper, narrative-level meaning[36] as opposed to word-level semantics. Some example questions include: "While I was reading the text, I could easily picture the events in it taking place", "I found myself thinking of ways the text could have turned out differently" and "I was mentally involved in the text while reading it" (see SI: Supplemental Methods). Thus, transportation reflects the act of building, representing and engaging with a situation model, which we take as a self-reported index of deeply processing a narrative.

Results are reported as proportions, with 1 as maximal transportation. Participants in the Word-scrambled condition reported feeling the least transported while reading, with progressively more transportation in the Sentence-scrambled and Intact conditions [$M_{Word-scrambled} = 0.41$; $M_{Sentence-scrambled} = 0.52$; $M_{Intact} = 0.64$; Two-tailed, independent samples t-tests; Intact vs. Word-scrambled: $t(158) = 11.74$, $p < 0.001$, $d = 1.86$; Sentence vs. Word-scrambled: $t(158) = 4.89$, $p < 0.001$, $d = 0.77$; Intact vs. Sentence-scrambled: $t(158) = 6.32$, $p < 0.001$, $d = 1.00$; for additional details, see SI: Supplementary Note II and III].

Similarly, scrambling also impaired performance on a 24-item multiple choice comprehension test, with the Intact condition reporting the highest scores [$M_{Word-scrambled} = 0.45$; $M_{Sentence-scrambled} = 0.67$; $M_{Intact} = 0.83$; Two-tailed, independent samples t-tests; Intact vs. Word-scrambled: $t(158) = 14.65$, $p < 0.001$, $d = 2.32$; Sentence vs. Word-scrambled: $t(158) = 9.87$, $p < 0.001$, $d = 1.56$; Intact vs. Sentence-scrambled: $t(158) = 6.33$, $p < 0.001$, $d = 1.00$; for additional details, see SI: Supplementary Note II and Fig. S1]. Therefore, scrambling the story indeed limited the extent to which participants were able to engage with the deep, situation-level meaning of the text.

### Stories elicit a lasting influence on spontaneous thought

Do stories linger in our minds and shape our spontaneous thoughts, more so than incoherent sequences of words and sentences? To answer this question, we examined participants' self-report of lingering. At the end of the experiment (~10 min later), participants indicated the extent to which the text continued to linger in their mind, using a scale of 1 (Not At All) to 7 (Very Much) (Fig. 1C). We found that self-reported lingering depended on the narrative coherence of the stimulus (Kruskal-Wallis rank sum test of Condition [Intact/Sentence-scrambled/Word-scrambled], Condition: $\chi^2(2) = 56.12$, $p < 0.001$, $\varepsilon^2_{ranked} = 0.23$). Participants who read the Intact story reported the strongest sense of lingering, significantly higher than those in the Sentence-scrambled [$Median_{Intact} = 5$, $Median_{Sentence-scrambled} = 4$; Two-

tailed Mann-Whitney U test: $U = 2207.5$, $p < 0.001$, $r_{ranked-biserial} = 0.31$] or Word-scrambled conditions [$Median_{Word-scrambled} = 2$; $U = 5259$, $p < 0.001$, $r_{rb} = 0.64$]. Participants who read the Sentence-scrambled version of the story, which still maintained some of its coherence, also reported a stronger sense of lingering than participants in the Word-scrambled condition [$U = 4560.5$, $p < 0.001$, $r_{rb} = 0.43$]. Although participants read all of the same words across all conditions, they reported more lingering as the objective situation-level coherence of their reading material increased.

Lingering was usually involuntary. In open-ended descriptions of their experience, participants often described lingering with an unintentional quality, distinguishing it from volitional rehearsal (e.g., "In the first round, the words I typed were considerably more organic than those in the second round, as I could not really get the story out of my head after reading it, so many of the associations were related to extraneous thoughts or associations with the story itself"). In a separate sample, we directly asked participants to describe the volitional quality of lingering: 51% described it as unintentional, only 7% as intentional, 18% as both, with the remaining 24% of participants describing it as neither or unsure (see SI: Supplementary Note XI; for all open-ended descriptions, see[37]).

Given that participants reported coherent narratives lingered in their minds, we reasoned that this lingering should bias their spontaneous thought and could manifest in their free-association data. To test this, we used document classification[33] to measure the difference in the content of free association chains generated pre- vs. post-story (for details, see Methods). In brief, for each condition (Intact, Sentence-scrambled and Word-scrambled) we trained a linear support vector machine classifier to predict whether a given free association chain was generated before or after reading. The input to the model was a vector of word counts from a single word chain, indicating the number of times each unique word from all free association chains was mentioned in that chain. The output of the model was a binary prediction of whether the word chain was "pre-story" or "post-story". Classification accuracy was the proportion of correct classifications across all free association chains (chance level = 50%; for details on null distribution, see Fig. 1D). If stories linger in a manner that reliably affects the content of free association, then the classifier should be able to discriminate between pre- and post-story chains, and this effect should be larger for a more coherent narrative.

Consistent with our predictions, the classifier was able to discriminate between pre- and post-story chains above chance for participants in the Intact (68% classification accuracy, One-tailed permutation test: $p < 0.002$) and Sentence-scrambled conditions (68% accuracy, $p < 0.002$), but not the Word-scrambled condition (52% accuracy, $p = 0.34$) (Fig. 1D). Therefore, both subjective measures (self-reported lingering) and objective measures (changes in the content of free association chains) indicate that narrative information lingers, and that this effect far exceeds what is elicited by decontextualized words.

Although, document classification did not differ across the Intact and Sentence-scrambled conditions, we next present evidence consistent with our hypothesis and participants' self-report data, indicating that coherent stories are more reliable drivers of lingering mental contexts.

### Stories linger more than sentences

Reading a series of words, without any overarching coherence, did not produce a lasting influence on spontaneous thought. However, document classification accuracy was similar for both the Intact and Sentence-scrambled versions of the Carver story, suggesting that sentence-level coherence may be sufficient to produce a lasting effect on spontaneous thought. To further examine whether Intact or Sentence-scrambled stories differ in the extent to which they linger in our minds, we collected three further datasets of the Intact and Sentence-scrambled manipulations: (I) a replication of the original

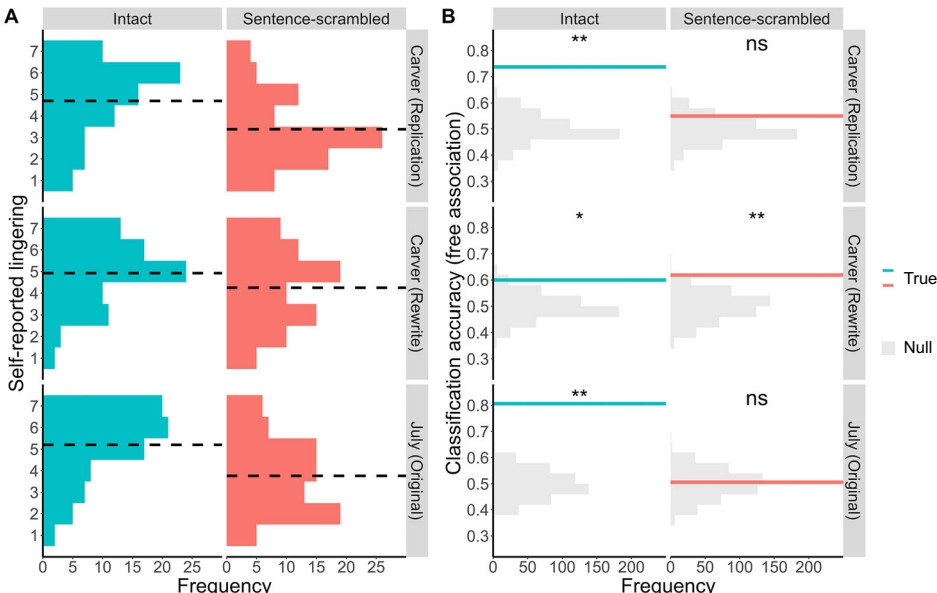

**Fig. 2 | Stories, not sentences, elicit a lasting influence on spontaneous thought. A** Histograms of participant responses, across three separate experiments, to the question: "To what extent did the text linger in your mind after reading it?". Participants provided their rating on a 7-pt scale: 7 = very much, 1 = not at all. Black dashed line represents the mean rating per condition. *n* = 80 participants per condition, per experiment. **B** Histograms of how accurately a document classifier could discriminate between pre- and post-story free association across three separate datasets. Classifiers were trained within-condition per dataset (*n* = 80 participants), using a leave-one-participant-out cross-validation procedure with 500 bootstraps. Solid lines represent the mean classification accuracy. Null distributions were estimated by randomly shuffling the labels of the held-out participant's word chains (pre, post) and recalculating classification accuracy over 500 permutations. Likelihood of achieving mean classification from the null distribution was calculated using a one-sided permutation test [*ns p* > .05; * *p* = <.05; ** *p* = <.01; Note 1: all *p*s are uncorrected with respect to multiple comparisons; Note 2: minimum *p* value estimate for this analysis is *p* < 0.002]. The exact accuracies and *p* values for each dataset and condition are as follows: *Carver-Replication* – Intact [74% accuracy, *p* < 0.002], Sentence-scrambled [55% accuracy, *p* = 0.130]; *Carver-Rewrite* – Intact [60% accuracy, *p* = 0.036], Sentence-scrambled [62% accuracy, *p* = 0.004]; *July* – Intact [81% accuracy, p < 0.002], Sentence-scrambled [51% accuracy, *p* = 0.420]. Source data for all panels are provided as a Source Data file.

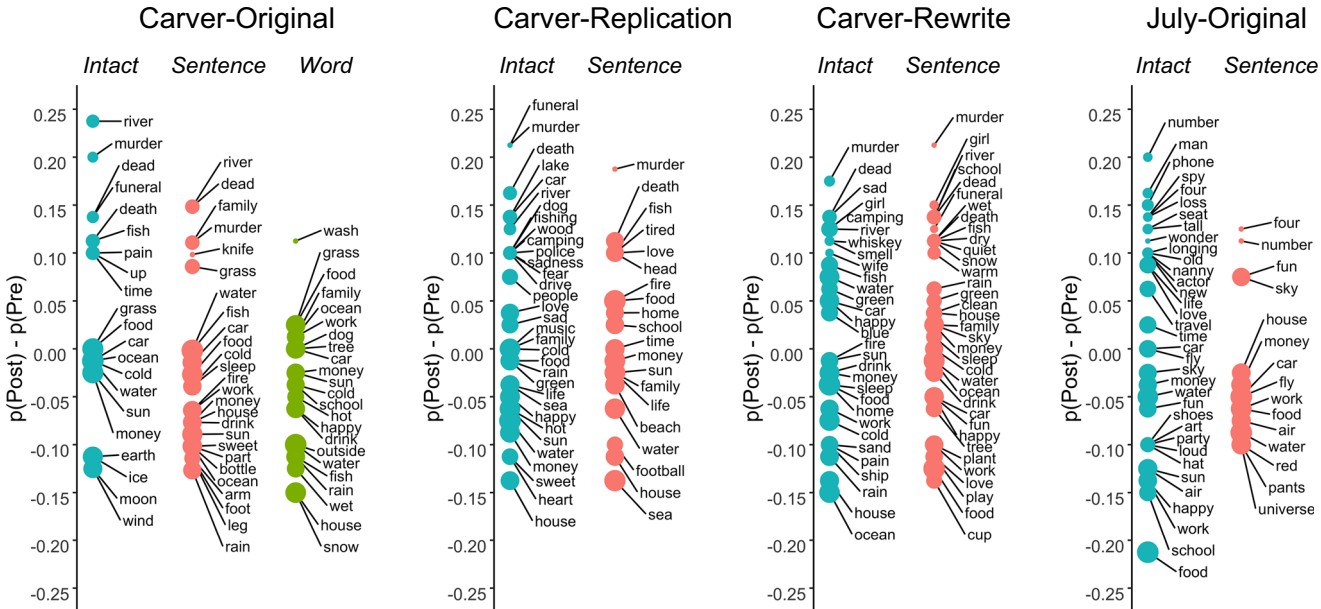

**Fig. 3 | Biases in free association content.** Bias in free association content from pre- to post-story is plotted for each dataset and condition in Experiment 1. Bias was defined as the proportion of post-story free association chains that contained a given word [p(Post)] minus the proportion of pre-story free association chains containing the same word [p(Pre)]. p(Post) and p(Pre) were both calculated separately for participants in a given condition and dataset, and thus were calculated based on the total of 80 free association chains. Therefore, positive values reflect words that are more likely to occur in post-story free association as compared to pre-story. Negative values reflect words that are more likely to occur in pre-story free association as compared to post-story. For legibility, only free associates that occurred in at least 16% of free association chains or showed a 10% bias for pre- or post-story are displayed. Size of points represents a given word's p(Pre). Source data for all panels are provided as a Source Data file.

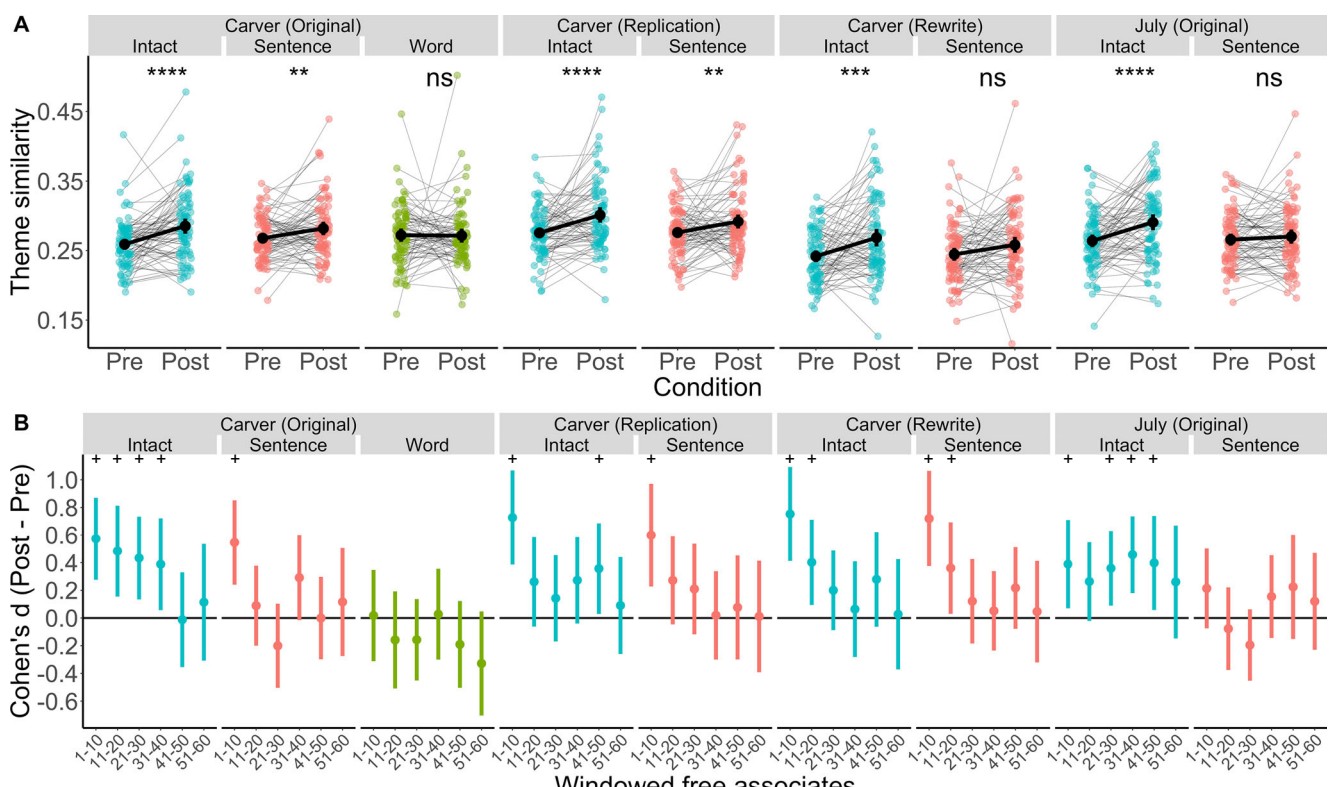

**Fig. 4 | Word embeddings reveal a lasting influence of story themes on post-story free association, especially after reading an intact story. A** Theme similarity pre- and post-story across all experiments. Theme similarity was averaged, per participant, across all associates produced pre- and poststory. Grey lines show the change in theme similarity within-participant. Group means are displayed using black circles. Error bars represent 95% confidence intervals. For display purposes, significance was estimated with uncorrected two-sided paired-sample t-tests comparing pre- vs. poststory theme similarity (see Supplementary Note XVI). Note that points in panel A were randomly jittered by .15 on the X axis to reduce overlap

and improve legibility. n = 80 participants per condition, per experiment. [ns $p > 0.05$; * $p = < 0.05$; ** $p = < 0.01$, *** $p = < 0.001$, **** $p = < 0.0001$]. **B** Timecourse of post-story theme similarity displayed using 10-associate windows. Effect size was calculated using Cohen's $d$, comparing theme similarity post-story minus pre-story within each window, and represented with a solid circle. Error bars represent 95% confidence intervals. Statistically significant effects ($p < 0.05$; as defined by 95% confidence intervals that do not include 0, without correction for multiple comparisons) are denoted with a + . n = 80 participants per condition, per experiment. Source data for all panels are provided as a Source Data file.

Carver story (Carver-Replication); (II) a rewrite of the Carver story, conveying the same plot using different words (Carver-Rewrite); and (III) an entirely different story, 'Roy Spivey' by Miranda July (July).

Within the three new datasets ($n$ = 160 per story, with $n$ = 80 per condition), the extent of self-reported lingering was again reduced by scrambling (Kruskal-Wallis rank sum test of Condition [Intact/Sentence-scrambled], Condition: $\chi^2(1)$ = 51.75, $p < 0.001$, $\varepsilon^2_{ranked}$ = 0.11; Story: $\chi^2(2)$ = 8.19, $p = 0.017$, $\varepsilon^2_{ranked}$ = 0.02; Fig. 2A). Once again, participants who read the Intact story reported a stronger sense of lingering than those who read the Sentence-scrambled version [*Carver-Replication: Median$_{Intact}$* = 5, *Median$_{Sentence-scrambled}$* = 3; Two-tailed Mann-Whitney U test: $U$ = 1828, $p < 0.001$, $r_{rb}$ = 0.43; *Carver-Rewrite: Median$_{Intact}$* = 5, *Median$_{Sentence-scrambled}$* = 4.5; $U$ = 2493, $p$ = 0.012, $r_{rb}$ = 0.22; *July: Median$_{Intact}$* = 6, *Median$_{Sentence-scrambled}$* = 4; $U$ = 1721, $p < 0.001$, $r_{rb}$ = 0.46], indicating that coherence at the sentence-level fails to elicit a sense of lingering to the same extent as an intact narrative (for comprehension test and transportation data, see SI: Supplementary Note II). Moreover, support vector machine classifiers trained on free association data from the Intact condition were able to predict whether a chain was produced pre- or post-story above chance for all stories (*Carver-Replication*: 74% accuracy, One-tailed permutation test: $p < 0.002$; *Carver-Rewrite*: 60% accuracy, $p = 0.036$; *July*: 81% accuracy, $p < 0.002$). Classifiers trained on free association data from the Sentence-scrambled conditions, however, only exceeded chance performance for one story only (*Carver-Replication*: 55% accuracy, $p = 0.130$; *Carver-Rewrite*: 62% accuracy, $p = 0.004$; *July*: 51% accuracy,

$p = 0.420$) (Fig. 2B). Thus, we found that, across four independent datasets, coherent narratives influenced the contents of subsequent thought more reliably than sentences or words.

### Story themes are upregulated in post-story free association

Across all four datasets (Carver, Carver-Replication, Carver-Rewrite, July), document classifiers could discriminate pre- and poststory free association chains – but what was changing in the free associations? We visualized the difference between pre- and post-reading patterns by calculating a bias score for each unique free associate. The bias score for each word measured the difference in the probability a free associate occurs in a chain before and after the story (Fig. 3).

Words that occurred more often in post-story free association, across-participants, reflected story content (Fig. 3). For example, the Carver story was centered around the discovery of the body of a young woman and the narrator's suspicion that her husband, who found the body on a camping trip, may have committed murder. Participants in the Intact condition from the Carver dataset were more likely to produce words such as "river", "murder", "dead" and "funeral" post-story as compared to participants in the Sentence or Word-scrambled conditions, even though each condition was composed of the exact same words. Associates related to murder and death were also more prominent post-story for Carver-Replication and Carver-Rewrite, which employed the same story. In the July dataset, where participants read a story about a chance romantic encounter between a celebrity and the narrator, words like "number", "man", "phone", "loss", "spy", and "four"

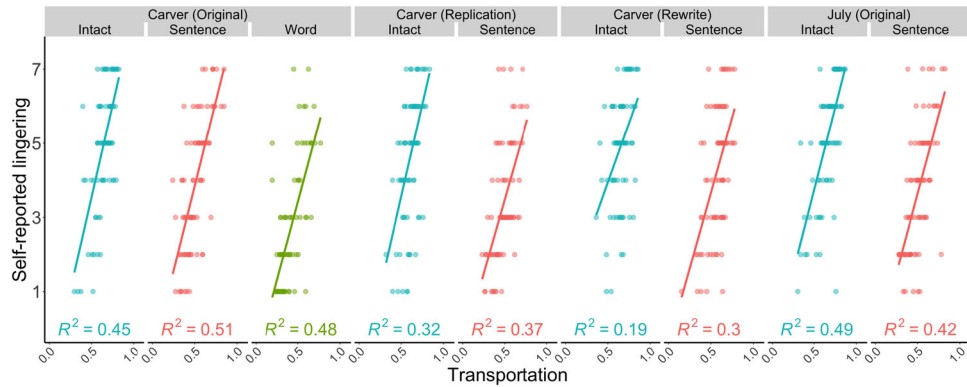

**Fig. 5 | Transportation into the story world predicts its lasting influence.** Scatterplots display the relationship between participant-level measures of self-reported transportation (Green & Brock, 2000) and self-reported lingering The colour of each point represents the experimental condition: blue = Intact, red = Sentence-scrambled and green = Word-scrambled. Transportation predicted lingering even in the Sentence-scrambled and Word-scrambled conditions, highlighting the importance of an individuals' own immersion in the content, irrespective of what the content is, in predicting its lasting influence on thought. $n = 80$ participants per condition, per experiment. Source data are provided as a Source Data file.

were more prominent post-story. These words relate to the story's plot, in which a celebrity (a famous actor in spy movies) shared his phone number with the narrator, withholding a single digit ("4") that he asked she commit to memory.

Both general themes and specific episodic content lingered substantially. Interpreting changes in bias as odds ratios [i.e., p(Post)/p(Pre)], we observed large odds ratios for both general theme words and more detailed content. For example, general themes such as "murder" (odds ratio = 6.3) and "loss" (odds ratio = 12) emerged in the Carver-Original and July stories, respectively. As examples of detailed content, "funeral" exhibited an odds ratio of 12 for Carver-Original, while "four" (odds ratio = 6.5) and "spy" (odds ratio = 6.5) emerged in the July story. Concretely, this means that 1 in 6 participants generated the word "funeral" after reading the intact version of the Carver story, while only 1 in 80 generated the word before the story. Similarly, more than 1 in 5 participants generated the word "four" after the intact version of the July story, compared to 1 in 40 who did so before. Thus, the lingering material includes both general themes and more specific episodic content, and the lingering is strong enough to be practically detectable in a group of people all exposed to a common narrative. For more examples of odds ratios, see SI: Supplementary Note XIII.

Next, we directly measured the persistence of story content in post-story spontaneous thought using a semantic analysis based on word embeddings (Global Vector embeddings; GloVe[34]). We quantified the semantic similarity between a participant's free association chains and the core themes of the story using a metric we defined as "theme similarity": the maximum cosine similarity between a given free associate and each of the story's 10 theme words. To derive these themes, participants generated 10 words that they believed related to the central themes and ideas of the text they had read, immediately following post-story free association. For each story, we selected the 10 theme words that were mentioned most frequently across participants (for details, see Methods). We then converted each free associate and each story theme to a 300-dimensional vector using the GloVe embeddings, allowing us to estimate the semantic similarity between words. Thus, we could measure the average theme similarity of free associates generated before and after reading each story (Fig. 4A).

Coherent stories were most likely to elicit a lingering effect in which their themes shaped post-story thought (Fig. 4A). Across all four datasets, the change in theme similarity from pre-story to post-story covaried with the level of scrambling of the text (Three-way ANOVA of Phase [Pre/Post], Condition [Intact/Sentence-scrambled], and Story [Carver/Carver-Replication/Carver-Rewrite/July]; Phase * Condition: $F(1,632) = 11.20$, $p < 0.001$, $\eta^2_G = 0.007$; for additional control analyses, see SI: Supplementary Note IV). Participants

showed more theme similarity post-story for both the Intact and Sentence-scrambled conditions, but the effect size was twice as large for the Intact condition [Two-tailed, paired samples t-test: Intact: $Pre = 0.260$, $Post = 0.286$, $t(319) = 8.42$, $p < 0.001$, $d = 0.56$; Sentence-scrambled: $Pre = 0.263$, $Post = 0.275$, $t(319) = 4.06$, $p < 0.001$, $d = 0.28$]. A separate paired t-test further confirmed no difference in theme similarity between pre- to post-story when the narrative was scrambled at the word-level [Word-scrambled: $Pre = 0.272$, $Post = 0.271$, $t(79) = -0.08$, $p = 0.930$, $d = -0.01$]. Finally, we also confirmed that the difference between post-story and pre-story theme similarity was positively correlated with self-reported lingering across all datasets [$r = 0.25$, $t(718) = 6.97$, $p < 0.001$]. For additional classification-based analyses supporting the presence of story content in post-story free association, see Supplementary Note XV.

### Stories linger for longer than sentences or words

Next, we sought to examine the time course of lingering. To this end, we divided free association chains into sequences of 10-word bins and calculated theme similarity for each bin separately. Given the average across-dataset production time for a single free associate was 4888 ms, a 10-word bin is approximately 49 s of the full 5-min of free association. The difference between post- and pre-story theme similarity was then represented as a Cohen's d effect size with 95% confidence intervals. Overall, the difference between post and pre-story theme similarity was highest immediately after the story ended, and the effect tended to persist over more free associates, particularly when the story was coherent. Measuring the number of 10-associate bins in which the post minus pre-story Cohen's d value was different from 0 (Two-tailed, one-sample t-test; $p < 0.05$, uncorrected) and averaging across experiments, we found that we could detect lingering story themes for approximately 3 bins (~147 s) for the Intact condition, 1 bin (~49 s) for Sentence-scrambled and 0 bins (~0 s) for Word-scrambled (Fig. 4B). Therefore, story themes tended to persist for longer into post-story free association after reading coherent stories, as compared to their constituent sentences and words.

### Transporting stories linger

Why do coherent stories linger in our minds? One possibility is that stimulus-level differences (e.g., objective coherence of the text) determine the depth of processing, which in turn predicts lingering. Specifically, coherent stories may contain types of information that are absent from scrambled sequences of words or sentences (e.g., agents, actions, intentions, embedded in broader situations evolving over time), and engaging with this kind of deep, situation-level meaning increases the likelihood of these thoughts persisting in mind.

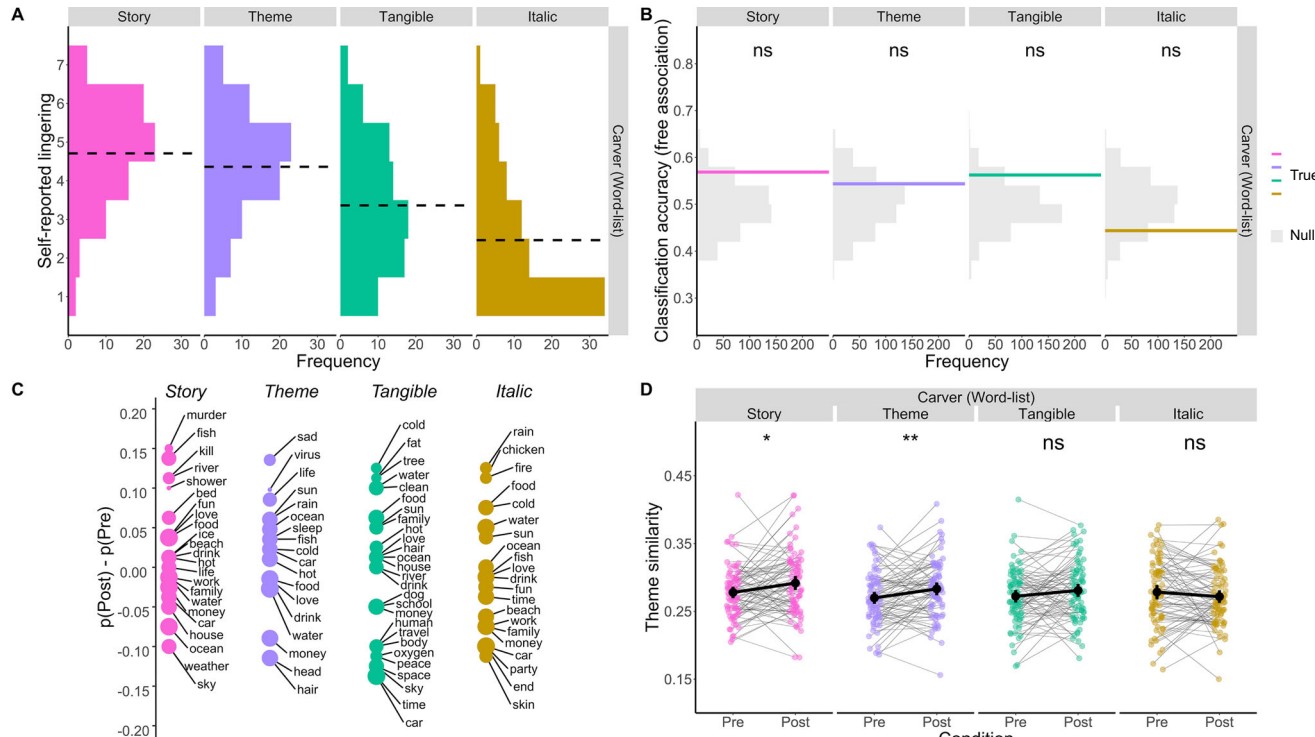

**Fig. 6 | Lingering following a list of words. A** Histograms revealed participants' self-reported lingering increased with deeper processing of a non-narrative stimulus (a word-list). Participants provided their rating on a 7-pt scale: 7 = very much, 1 = not at all. Black dashed line represents the mean rating per condition. $n = 80$ participants per condition. **B** Results of document classification, however, show no evidence of a detectable change in words used in pre- vs. post-task free association. Classifiers were trained within-condition per experiment ($n = 80$ participants), using a leave-one-participant-out cross-validation procedure with 500 bootstraps. Solid line represents the mean classification accuracy. Null distributions are plotted in gray. The exact accuracies and p-values for each of the one-tailed permutation tests is as follows: Story [57% accuracy, $p = 0.056$]; Theme [54% accuracy, $p = 0.176$]; Tangible [56% accuracy, $p = 0.072$]; Italic [44% accuracy, $p = 0.842$]. **C** Bias in free association content from pre- to post-task is plotted for each condition. Bias was defined as the proportion of post-task free association chains that contained a given word [p(Post)] minus the proportion of pre-task free association chains containing the same word [p(Pre)]. p(Post) and p(Pre) were both based on the total of 80 free association chains per condition. For legibility, only free associates that occurred in at least 16% of free association chains or showed a 10% bias for pre- or post-task are displayed. Size of points represents a given word's p(Pre). **D** Theme similarity pre- and post-task highlights some evidence for an increase in similarity to list themes after deeply processing the word list. Grey lines show the change in theme similarity within-participant. Group means are displayed using black circles. Error bars represent 95% confidence intervals. $n = 80$ participants per condition. For display purposes, significance was estimated with uncorrected two-sided paired-sample t-tests comparing pre- vs. post-task theme similarity [ns $p > 0.05$; * $p =< 0.05$; ** $p = <0.01$, *** $p = <0.001$, **** $p = <0.0001$]. For additional details regarding paired t-tests, see Supplementary Note XVI. Source data for all panels are provided as a Source Data file.

Alternatively, lingering may be better explained by an individual's appraisal of a stimulus, rather than the properties of the stimulus itself. According to this hypothesis, if an individual is able to extract, represent and immerse themselves in the world of a story, it should linger in mind, regardless of the text's objective coherence. In order to adjudicate between these two possibilities, we examined whether participant-level narrative transportation ratings, a proxy for the depth with which an individual engaged with the text, could predict subjective and objective measures of lingering, independently of whether the text was intact or scrambled. To this end, we included transportation in a 10-fold cross-validated regression model with backwards stepwise feature selection (see Methods). Regression models included participant-level measures of transportation; performance on a comprehension test (i.e., memory for specific verifiable details from the story); and experimental condition (i.e., Intact / Sentence-scrambled / Word-scrambled). If lingering is determined by the properties of the text itself, the experimental condition should be the best predictor. If lingering instead results from the depth with which an individual engages with the text, irrespective of its objective coherence, transportation should outperform all other predictors.

An individual's sense of transportation was a better predictor of their lingering experience than the objective coherence of the text they read (Fig. 5). When predicting self-reported lingering, the final model contained two variables, in which transportation accounted for the bulk of the variance, with an additional contribution from comprehension test performance [final model: $R^2 = 0.51$; $F(2,1437) = 739.6$, $p < 0.001$; transportation: $b = 0.669$, $t(1437) = 31.92$, $p < 0.001$; comprehension: $b = 0.085$, $t(1437) = 4.04$, $p < 0.001$]. When predicting the difference between post and pre-story theme similarity, the final model only contained transportation [final model: $R^2 = 0.009$; $F(1,1438) = 5.14$, $p = 0.024$; transportation: $b = 0.06$, $t(1438) = 2.3$, $p = 0.024$]. Thus, the extent to which an individual felt transported into the story world was an important determinant of post-story lingering, explaining more out-of-sample variance than their experimental condition (Intact/Scrambled) or their comprehension of verifiable story details (also see Supplementary Note XIV).

## Deeply-processed words linger

The extent to which a participant felt transported while reading a text predicted the likelihood of the story themes lingering in their spontaneous thought. Interestingly, performance on a comprehension test of verifiable story details did not predict lingering as strongly. This is consistent with our hypothesis that lingering is not simply a function of whether participants encoded the objective features of a stimulus, but crucially depends on the depth of processing they employed. Encoding a given experience can entail attending to its surface-level features

(e.g., the verbatim text in a written story) to the broader meaning it represents (e.g., constructing a situation model from the written text[29,36]), and the more likely we are to attend to its deeper meaning, the more likely it will linger in our minds (for related ideas, see[26]). In fact, when participants perform a cover task that encourages shallow processing, lingering is reduced, despite the text itself being objectively coherent (see SI: Supplementary Note IX – Experiment 4).

If deeper processing drives lingering, then stories should not be alone in their propensity to linger. Instead, the content of any text should linger if participants attempt to create overarching situation model. Therefore, we conducted an additional experiment in which participants encoded a fixed list of words, and we manipulated the depth of processing that they applied during encoding. Thus, rather than scrambling a story to reduce situation-level coherence, we presented participants with a non-narrative stimulus and manipulated the depth with which they engaged with it. Three hundred and twenty participants were exposed to a list of 268 words (of which 201 words were related to ideas and characters from the Carver story, and the remaining words chosen to be distinctive yet unrelated to the story, see Methods) while performing one of four cover tasks. Cover tasks ranged from orienting participants towards the surface-level features of the word-list (i.e., deciding whether or not a word was italicized), to the word-level meaning (i.e., deciding whether or not a word represented something tangible) or the list-level meaning (i.e., deciding whether or not a word belonged to a latent theme or story embedded in the word-list; see Methods). Words were ordered so that the main characters, locations and events were mentioned early, allowing participants to get a sense of the list-level meaning if they were instructed to seek it. For example, the first 15 words in the list were as follows: "Claire", "Stuart", "couple", "small-town", "Stuart", "buddies", "camping", "fishing", "find", "girl", "dead", "Claire", "suspicion", "mistrust", "murder". Unrelated decoy words were selected to have high positive valence, for example: "cheerful", "bliss", "luxury", "peaceful", "happy", "magical". Decoy words were pseudorandomly interspersed within the list with an average of 6.01 (SD = 3.47) story words between consecutive decoys (for full list, see[37]).

Although all participants saw an identical list of words, and although they were all encoding the words as confirmed by their cover-task performance (see SI: Supplementary Note V, VI and VII), they reported very different levels of lingering (Kruskal-Wallis rank sum test of Condition [Italic/Tangible/Theme/Story], Condition: $\chi^2(3) = 78.57$, $p < 0.001$, $\varepsilon^2_{ranked} = 0.25$; Fig. 6A). Participants in the Story and Theme conditions reported the strongest sense of post-task lingering ($Median_{Story} = 5$, $Median_{Theme} = 4.5$), followed by Tangible and Italic ($Median_{Tangible} = 3$, $Median_{Italic} = 2$) [Two-tailed Mann-Whitney U test: Story vs. Theme: $U = 3604.5$, $p = 0.12$, $r_{rb} = 0.14$; Story vs. Tangible: $U = 4659$, $p < 0.001$, $r_{rb} = 0.47$; Story vs. Italic: $U = 1038$, $p < 0.001$, $r_{rb} = 0.67$; Theme vs. Tangible: $U = 2054$, $p < 0.001$, $r_{rb} = 0.36$; Theme vs. Italic: $U = 1293.5$, $p < 0.001$, $r_{rb} = 0.60$; Tangible vs. Italic: $U = 2124$, $p < 0.001$, $r_{rb} = 0.34$]. Therefore, participants who were instructed to process the words deeply (i.e., Story and Theme conditions) were more likely to report stronger post-task lingering.

Although processing depth was strongly associated with self-reported lingering, there was relatively weak objective evidence of lingering in post-story free associates. Support vector machine classifiers failed to predict whether a chain was produced pre- or post-task above chance for all conditions (Story: 57% accuracy, One-tailed permutation test: $p = 0.056$; Theme: 54% accuracy, $p = 0.176$; Tangible: 56% accuracy, $p = 0.072$; Italic: 44% accuracy, $p = 0.842$; Fig. 6B). Although data-driven document classifiers failed to detect any reliable changes in free association pre- vs. post-task, plotting bias scores did reveal that some story-related words (e.g., "murder") were more likely to occur post-task. This was specifically true for participants in the Story condition, who were required to infer the events of a latent story from the word list (Fig. 6C). This was further corroborated when

examining theme similarity from word-embeddings (Fig. 6D), which revealed a weak, though significant, interaction between task condition and phase [2-way ANOVA of Phase [Pre/Post] and Condition [Italic/Tangible/Theme/Story]; Phase * Condition: $F(3,316) = 3.40$, $p = 0.020$, $\eta^2_G = 0.009$]. Paired t-tests indicated that participants in both the Story and Theme conditions showed more theme similarity post-task as compared to pre-, which was not true for participants in the Italic or Tangible conditions [Two-tailed, paired sample t-tests: Italic: $Pre = 0.278$, $Post = 0.272$, $t(79) = -1.30$, $p = 0.200$, $d = -0.15$; Tangible: $Pre = 0.272$, $Post = 0.281$, $t(79) = 1.67$, $p = 0.100$, $d = 0.20$; Theme: $Pre = 0.270$, $Post = 0.283$, $t(79) = 4.06$, $p = 0.009$, $d = 0.32$; Story: $Pre = 0.278$, $Post = 0.292$, $t(79) = 2.51$, $p = 0.014$, $d = 0.32$]. Although the effect sizes reported here are smaller than those we observed with Intact stories ($d = 0.56$), attending to the latent across-element meaning in a list of words also resulted in a lasting mental context.

## Discussion

What determines whether a past experience persists in our mind? Despite the prominence of history-dependence in our models of the human mind and memory[5,6,8,38,39], we have little empirical evidence to support our intuitive sense that meaningful experiences resonate with us, shaping our thoughts in the minutes after they end. Here, we empirically demonstrate that, when people interpret a text as a narrative of interconnected situations, instead of focusing on the semantic or perceptual features of individual sentences or words, they experience a lingering influence on the trajectory of their subsequent thoughts for minutes afterward.

Participants who read short stories reported that the text lingered in their minds for several minutes, but this effect was greatly reduced amongst those who read the same stories with sentences or words in a shuffled order (Fig. 1C & 2A). Although participants' experiences of lingering varied, many described the ongoing mental presence of the text as something unbidden or even distracting: "I think maybe the story stayed with me and affected me a little. I tried to not let it influence me and to go where the words took me". In fact, when we asked a separate group of participants to describe the volitional quality of their lingering experience, the majority of them confirmed that it was unintentional (see SI: Supplementary Note XI). Thus, the lingering phenomenon appears to be distinct from intentional rehearsal or the kinds of explicit memory that typically benefit from deep processing[26], but instead acts as a kind of latent constraint[5,8] on participants' spontaneous thought.

Participants' free association chains were altered after reading the coherent story (Fig. 1D & 2B) and their words were semantically closer to the story's themes (Fig. 3 & 4). Critically, these lingering influences were strongest amongst participants who reported being transported into the story world, arguably reflective of deep processing[36], regardless of the objective coherence of the text they read (Fig. 5). We further generalized these observations to a case in which all participants read the same word list, but only some of them sought out a latent story or theme within the list (Fig. 6). Overall, these data indicate that information will persist in our thinking when we seek to extract and represent its deeper situation-level meaning.

The extent to which a past experience lingered in a participant's mind was strongly dependent on whether they encoded it deeply, as a situation. A relationship between history-dependence and situation-level meaning can also be seen when imaging the brain while engaging with narratives. For example, functional magnetic resonance imaging (fMRI) studies reveal pronounced history-dependence in higher-order association cortices (e.g., regions of the default mode network; DMN[40]), only when participants watch or listen to a coherent story[32,41,42]. Therefore, the way DMN regions respond to what is onscreen at a given moment in time depends on how the current event fits within the narrative context of what happened several minutes into the past[38,43,44]. The finding that deeply processed stories linger in the mind

(immediately after an experience) is also consistent with the behavioural finding that stories and situations persist in the form of lasting memories. Information presented in story form is better remembered than non-narrativized information[45], narrative coherence benefits memory for naturalistic events[46], and the act of studying a word-list as if it were a story markedly improves recall[47] (for a review, see[48]). Beyond improving memory, stories also have lasting consequences on how we behave and what we believe[49–52]. Other examples of situation-level information, like social interactions and emotions, also share this sticky or enduring quality: persisting in our thoughts[19–21] and shaping how we learn and remember[22–25]. Critically, by using free association and a story-scrambling procedure, we were able to directly quantify this lingering phenomenon in spontaneous thought while relating it to one's ability to extract situation-level meaning.

Why should attending to situation-level meaning elicit a lasting mental context? A potential explanation comes from the levels of the processing framework of human memory[26,53]. The levels of processing framework stems from work on perception, where the perceptual process was conceptualized as a hierarchical series of tests at different levels of analysis[54]. Early levels are concerned with the physical properties of a stimulus, while later levels examine more abstract stimulus properties such as meaning and implication. According to levels of processing, the persistence of the stimulus in memory is a function of these levels of analysis: stimuli that are processed at later (deeper) levels are more likely to form lasting representations in memory. For example, studying a word list by deciding whether or not each word is capitalized results in poorer recognition than studying them based on their fit in a sentence[55]. However, the depth of processing does not stop at word-level semantics. We contend that engaging with the situation-level meaning of a text is a prime example of even deeper meaning-centered processing[36] and should result in persistent representations in memory. While levels of processing models traditionally concern persistence in memory rather than spontaneous thought, there is reason to consider these constructs are related. For example, overall memory performance is positively correlated with history-dependence (i.e., temporal clustering) in freely recalled word lists[56]. Also, the extent to which a recent social experience permeated thoughts during a post-task rest period predicts subsequent memory for the original experience[19]. Lingering in spontaneous thought may be a natural consequence (or antecedent) of robustly encoded memories.

Precisely how deep processing results in lingering remains an open question[53]. If we consider all of the individual units of our knowledge and experience as nodes on a graph, deep processing may be operationalized as a learning function that results in lasting increases in the edge weights between the nodes of an input and the nodes of related knowledge and experiences. If we then model spontaneous thought as a random walk on such a network[57], we should be more likely to traverse these deeply processed edges again (i.e., lingering). Furthermore, the consequences of deep processing according to this model would not be limited to non-volitional random walks, but would also benefit more rule-based search processes on the network, consistent with the well-documented benefits of deep processing on tests of explicit memory[26]. However, how a pure associative network model could support the lingering of more complex mental representations, such as our current concerns or goals[58] remains unclear. Agent-centered models[59,60] that combine decisional and episodic memory processes may be necessary to capture the real-world phenomenon in which our thoughts during one task or interaction persist for minutes into the next task or interaction. In the brain, deep processing may result in lingering via a propensity to drive activity in higher-order association cortices (e.g., regions of the DMN)[38,39]. Higher-order association cortices possess distinctively slow-drifting intrinsic dynamics, likely due to their elevated levels of local-circuit and inter-regional recurrence[61,62]. Thus, if deep processing especially

involves these brain regions, they are well-placed to generate lasting neural reverberations and, perhaps, lingering mental contexts.

The notion that parsing situation-level meaning is an example of deep processing is consistent with hierarchical models of discourse comprehension. Kintsch (1998) conceptualized comprehension as a multilayered system, beginning with a "surface code" to provide a verbatim representation of a text's words and syntax, and ending with a "situation model" that summarizes the broader happenings they describe. The act of constructing and elaborating on a situation model requires a reader to move beyond the text itself and consider its deeper structure[36]. In line with levels of processing, deep processing resulted in a persistent representation: participants who were most likely to succeed in constructing a situation model (i.e., those in the Intact condition), were also most likely to show evidence of the text lingering in spontaneous thought (Fig. 2 & 3). Critically, our feature selection procedure indicated that a sense of transportation into the story world (i.e., the act of building, representing and engaging with a situation model from the text[36]) predicted lingering over and above the objective coherence of the text that was read (Fig. 5). Also, participants who read a list of words as if it were a story reported more lingering (Fig. 6). Therefore, the objective narrative coherence of the text mattered less than the extent to which an individual was able to deeply engage with it, via the construction of an immersive situation model. From this perspective, it is not surprising that our document classifier occasionally exceeded chance levels at predicting pre- vs. post-story free association from participants in the sentence-scrambled conditions (Fig. 1B & 2B). Transportation amongst participants who read the sentence-scrambled texts was not at floor (SI: Supplementary Note II; Fig. S2), which may result in group-level lingering, although less reliably than for a coherent story.

The construction of situation models is not the only route to a lasting mental context: participants also reported a comparable extent of lingering after judging whether each word in a list belonged to a common theme (Fig. 6). This "Theme" task did not explicitly require participants to encode the words in terms of situations, but did require that they carefully attend to how each word may be related to one another. In particular, they needed to examine words beyond their surface-level features in order and develop a latent theme over the course of the list. From the perspective of the levels of processing framework, then, participants in the Theme condition are still deeply processing the word list. Thus, it seems that situation-level thought (or narrative thinking) is not the only example of deep meaning-centered processing. Other kinds of deep thinking, concerning what a stimulus implies rather than its physical properties, should also form lasting mental contexts. Take the example of cognitive fixedness in problem solving[63,64]. After solving a series of problems using a complex algorithm, we often continue applying this unnecessarily complex solution even in the face of simpler problems. Luchins (1942) elicited this kind of lasting mental context using problems of measuring water volume with different-sized cups – a far cry from a story. Accordingly, one important feature of deep thinking could be a sense of immersion – where one is lost in the performance of a certain computation[65], whether it be performing a series of arithmetic steps or elaborating on the happenings in a story world. Therefore, the dimension that differentiate stories and situations from other paths to deep thinking may be our natural affinity for narrative information[66,67] as opposed to the possibility that stories themselves have an intrinsic propensity to linger in our minds.

Having considered which kinds of processing increase mental lingering, we must finally ask: why linger at all? In an ever-changing world, why should any of our experiences colour the trajectory of our thoughts for minutes after they end? One explanation may be that experiences that linger are better consolidated into memory. In rodents, hippocampal neural ensembles associated with recent experiences are spontaneously reactivated during sleep[68] and post-

task wakefulness[69] (for evidence in humans, see[70,71]). Critically, interrupting this 'hippocampal replay' impairs memory formation[72]. An intriguing possibility may be that lingering in spontaneous thought is a behavioural correlate of this hippocampal replay mechanism. Given our hypothesis that deep processing drives lingering, experiences associated with deep thinking may be preferentially replayed, in turn helping us prioritize significant or meaningful events in memory[73,74]. Furthermore, memory consolidation does not retain all details equally: idiosyncratic details tend to be lost while the central gist is preserved[75,76]. Perhaps the reverberation of overarching story themes in spontaneous thought may be a mechanism in which these central details are preferentially reactivated and strengthened in memory. Future studies examining the consequences of lingering on memory are necessary to test these ideas.

Philosophers and psychologists have noted that our stream of thought echoes recent and distant memories, and that each moment informs the meaning of the next[1,2]. Here we demonstrated that the extent of this history dependence is not a fixed parameter. Instead, the extent to which our recent past lingers into subsequent thought increases as a function of processing depth[26,53]. The more we consider the deep situation-level meaning of an experience, the more likely it will exert a lasting mental context and shape the trajectory of our subsequent thoughts.

## Limitations

While our data provide empirical evidence that aspects of the narrative-level meaning of a text can persist in our spontaneous thoughts, there are some important limitations. First, our free association task operates at the level of words, which is almost certainly not the level of representation of our spontaneous thoughts, nor are words likely to be what is lingering in mind after we read a story. Instead, introspection would suggest that when a story lingers in mind, what is persisting can be better described as a more complex kind of mental representation (e.g., events, situations, topics, themes, emotions; see SI: Supplementary Note XII). As a result, our objective measures of lingering (i.e., document classification and theme similarity) only indirectly reflect a subset of the actual subjective experience of lingering. Consistent with this point, the correlation between theme similarity (Post- minus Pre-story) and self-reported lingering was fairly small when calculated across all datasets in Experiment 1 ($r = 0.25$). We opted for a free word association task as the semantic associations between spontaneously generated words (i) are thought to sample the more complex mental constructs that form the basis of our thoughts[77] and (ii) are readily quantified using tools from natural language processing[34]. While our approach provides a reliable way to quantify lingering without self-report, future studies using more unconstrained "think-aloud" paradigms[78] and richer language embeddings grounded in agency[79] may still improve the correspondence between our objective measures and subjective experience. Another limitation of this study is the lack of an objective index of processing depth. This criticism has been leveled against the original levels of processing framework[80] and it limits the extent to which we can definitively determine that processing depth drives lingering. Nonetheless, the sensitivity of regions of the DMN to the situation-level meaning of narratives[32,38,42] provides a potential path for future studies to leverage brain activity as a quantitative index of deep processing.

## Methods

All research was approved by the Johns Hopkins University Homewood Institutional Review Board and informed consent was given by each participant before participating.

### Experiment 1: free association, pre- and poststory
**Experimental procedure.** Participants were recruited via Amazon Mechanical Turk (AMT) or Prolific. Data were collected using the

Psiturk platform[81,82] with four separate versions of the experiment (Carver, Carver-Replication, Carver-Rewrite and July; for details, see Narrative Stimuli). All stories comprised the same sections: (I) Math; (II) Pre-story free association; (III) Self-paced reading, (IV) Post-story free association; (V) Themes generation; (VI) Narrative transportation; (VII) Comprehension test; (VIII) Demographics and Strategy; (IX) Self-reported lingering. Additional methodological information can be found in the SI: Supplemental Methods.

**Free association.** Participants were introduced to a task called the "word chain game", in which they were asked to type any words that came to mind for a total of five minutes. The task consisted of a blank white screen with a cue word in the black font (e.g., WATER) and an empty field for text entry below it. The cue word remained onscreen for 2000 ms upon task onset and then faded away over 500 ms. Participants were instructed to type whatever words came to mind, as they came to mind, into the text entry field. The cue word acted as a starting point, to help participants begin generating their own free associations. Cue words were manually selected by the experimenter to be related to the story. Each story was associated with two cue words, one for pre-story free association and one for post, counterbalanced across participants (Carver/Carver - Rewrite: "water", "body"; July - "plane"; "secret"). After typing each word, participants were instructed to press enter, causing the word to disappear from the text field and reappear in the cue position for 500 ms before fading away entirely. This procedure ensured that participants did not have continued access to the words they had previously generated. This task designed to be freeform, with the only additional instruction being that participants should avoid stringing words into sentences.

For details regarding the number of words produced during free association and their composition in terms of parts of speech, see SI: Supplementary Note I. For a replication of our empirical results using free association with a cue word that is unrelated to the story (i.e., "Type a word to begin!"), see SI: Supplementary Note VIII – Experiment 3.

**Self-paced reading.** Participants progressed through the text at their own pace by pressing space bar after reading each sentence. All stories were between 2,158 to 2,798 words in length, ranging between 196 and 268 sentences.

**Theme generation.** Participants freely generated up to 10 words relating to the central themes and ideas of the text they read.

**Narrative transportation.** Participants completed a modified version of the Narrative Transportation Questionnaire[35], a 13-item scale assessing the extent to which participants were transported into the story while reading it (e.g., "While I was reading the text, I could easily picture the events in it taking place"; "I could picture myself in the scene of the events described in the text"; "The text affected me emotionally"). Participants responded to each item on a 7-point scale ranging from Not At All (1) to Very Much (7). All scores were summed and reported as proportions, where 1 is the highest achievable score of transportation. For additional scoring details, see SI: Supplemental Methods.

**Comprehension test.** Comprehension of verifiable story details was measured using a 26-item 4-alternative-forced-choice test. Questions were presented out of chronological order. 2 items were catch trials and the remaining 24 assessed comprehension of the intact story. Half of the content questions were very general (e.g., "Which of the following beverages figured most prominently in the passage?") while the remaining half were specific and plot-focused (e.g., "How did Claire's husband encounter the body?"). Questions for both stories can be found online[37].

**Self-reported lingering.** Following standard demographics questions and open-ended questions asking participants about the strategies they employed during the Math and Free association sections, participants were asked about their subjective experience of the text "lingering" in their minds. Specifically, participants were asked to (i) describe any differences they felt between pre- and post-story free association and (ii) provide a rating of their experience of the text lingering in their minds (i.e., "To what extent did the text linger in your mind after reading it?") on a scale of 1 (Not At All) to 7 (Very Much). Participants were further asked to "please describe any differences you may have felt between playing the word chain game before and after reading the text" with an open-ended response (for descriptions, see[37]).

**Narrative stimuli.** Participants read one of 3 stories: (1) *So Much Water So Close To Home* by Raymond Carver (Carver; Carver-Replication); (2) a rewrite of Carver, conveying the same narrative information using different words (Carver-Rewrite); and (3) *Roy Spivey* by Miranda July (July). These stories were chosen because they were easy to read (at a Grade 5 reading level, or below) and short (under 3000 words), yet still immersive and evocative. For additional details, see SI: Supplemental Methods.

**Story scrambling.** In the Intact condition, participants read each sentence, one at a time, in the order of the published story. In the Sentence-scrambled condition, participants read the identical sentences, however, the order of the sentences was randomly shuffled. In the Word-scrambled condition (specific to the original Carver story), the story was parsed into 5-sentence segments and the original sentences were then repopulated by randomly drawing the same number of words from all the words belonging to a segment. In this way, we created a document that contained the same words are the Intact story, in a similar (large-scale) order to the original text, while largely obscuring the overall meaning. Note that the shuffling procedure was applied to the stimulus once, and all participants in the scrambled conditions read the same Sentence- or Word-scrambled version of the story. Furthermore, we modified the self-paced reading task in the Word-scrambled task to ensure that participants read each word. To this end, we included 66 yes/no probe trials interspersed within the reading task (following a Poisson distribution with a mean of 4 sentences): during yes/no probe trials, the story text was replaced with a single word in red font below the question "Was this word in the previous sentence?". 50% of the probe trials were targets and 50% were foils. Target and foil words were manually selected by the experimenter to reflect a comparable distribution of parts of speech as the original text, ensuring participants would have to pay attention to all the words in each sentence to achieve above chance performance.

**Participants.** One thousand and twelve participants took part in Experiment 1 and were recruited via Amazon Mechanical Turk (versions: Carver, Carver-Rewrite, July) or Prolific (version: Carver-Replication). MTurk data were collected over the span of June 2019 – March 2020. Prolific data were collected during September 2020. The experiment lasted approximately 45 min. Participants were paid $6.00 USD for their participation and provided informed consent before participating.

After exclusions and quality-assurance checks (see SI: Supplemental Methods), a total of 720 participants were included in the final sample ($N_{male}$ = 360; $N_{female}$ = 354, with 6 participants selecting "None of the above / Prefer not to identify"). Median age range in the final sample was 35–39 years of age ($Q_2$ = 25–29, $Q_3$ = 45–49, min = 18–19, max = 70–74). Eighty participants were included in each condition, per story: Carver [Intact/Sentence-scrambled/Word-scrambled], Carver-Replication [Intact/Sentence-scrambled], Carver-Rewrite [Intact/Sentence-scrambled], and July [Intact/Sentence-scrambled].

**Document classification.** To determine whether pre- vs. poststory free association chains were statistically discriminable, we used a support vector machine (SVM, implemented in R[83]) to perform document classification. A document-term matrix was computed from all free association chains belonging to participants from a given condition, for a given story. Each row of the matrix represented a free association chain, with columns for every unique word from all free association chains, and the matrix values composed of each word's raw count in each chain. Each participant is associated with two rows in the document-term matrix: one for pre-story free association and another for post. Word counts were rescaled by mean centering each column (i.e., word) within-participant, and then dividing the full column by the across-participant standard deviation. Using a leave-one-participant-out cross-validation procedure, we trained a linear SVM to discriminate between pre- and post-story free association chains in the document-term matrix. Specifically, the input features to the model were the term-frequencies for a held-out participant's free association chain (i.e., a held-out row in from the document term matrix). The model output was a prediction as to whether the input was generated pre- or poststory. Accuracy was then computed by comparing the assigned binary labels against ground truth (i.e., true pre- and post-story labels). We repeated this analysis 500 times and report the mean as our estimate of true classification accuracy. Note that the model's predictions are based on whether or not a word was in a word chain and does not account for the order words appeared in free association.

To determine whether classification accuracy was above chance, we generated a null distribution of 500 accuracy values. The null accuracy values were generated using the same procedure as described above, after randomly shuffling the pre- post- labels from the test dataset for each fold in cross-validation procedure. We then computed the proportion of null accuracy values greater than the empirical classification accuracy.

**Word embeddings and "theme similarity".** To test whether story themes were present in post-story free association, we measured the similarity of vectors representing free associates and vectors representing the core story themes. Each associate and theme word was mapped onto a 300-dimensional vector (GloVe; version: Wikipedia 2014 + Gigaword 5[34]). Free associates without corresponding vectors in pretrained corpus were dropped from subsequent analyses. For each story, we defined "theme words" as the 10 words mentioned most frequently across participants during the theme generation task, collapsing across conditions. Theme words for each story were: Carver ["murder", "death", "funeral", "fishing", "girl", "family", "camping", "river", "beer", "sex"], Carver-Rewrite ["murder", "funeral", "wife", "husband", "death", "fishing", "camping", "suspicion", "mystery", "friends"], and July ["four", "celebrity", "plane", "airplane", "husband", "secret", "number", "affair", "actor", "famous"].

We quantified the similarity of each free associate to the story themes, before and after reading. To this end, we calculated a measure we refer to as "theme similarity". For the $n$-th free associate, represented by embedding vector $A_n$, the theme similarity was computed as the maximal cosine similarity across all theme words:

$$\text{theme similarity}(\mathbf{A_n}) = \max_i \left( \frac{\mathbf{A_n} \cdot \mathbf{B_i}}{\|\mathbf{A_n}\| \cdot \|\mathbf{B_i}\|} \right) \tag{1}$$

where $B_i$ is the embedding vector for the $i^{th}$ theme word. Theme similarity was calculated for every word in each free association chain and then averaged per chain.

Note that our two approaches for analyzing free association data (theme similarity and document classification) may not always coincide. While theme similarity makes use of a continuous estimate of semantic similarity between any two words available to the pretrained model[34], our implementation of document classification relies on

exact word matches across participants. Consider the following example: all participants in the document classifier's training set have the word "stream" in their post-story chains, while none have it in their pre-story chains. Also, the semantically-related "creek" was not used by any participants in the training set during pre- or post-story. Now, if a held-out participant uses the word "creek" post-story, but not "stream", the classifier will not be able to make use of the importance of the word "stream" to predict that "creek" should also imply the chain was generated post-story. However, in the context of theme similarity, instances of the words "stream" and "creek" will both show strong semantic similarity to the story theme "river", and thus "creek" will contribute signal to our theme similarity analysis but not to document classification.

Additional information and control analyses are reported in SI: Supplementary Note IV.

**Predicting lingering via stepwise feature selection.** To determine the measures that best predicted our subjective (self-reported lingering) and objective (theme similarity) measures of lingering, we performed stepwise feature selection. Predictors input into the regression model were: participant-level scores on transportation[35], participant-level performance on the 24-item multiple choice comprehension test, and experimental condition (i.e., Intact / Sentence-scrambled / Word-scrambled). Self-reported lingering and theme similarity ($M_{post-story} - M_{pre-story}$) were included as dependent variables, in two separate regression models. Stepwise feature selection was performed using a backwards stepwise linear regression with 10-fold cross-validation, implemented in R using caret (model = leapBackward)[84]. All numerical measures were z-scored prior to inclusion in the model.

## Experiment 2: depth-of-processing word-list variant

**Experimental procedure.** Procedures were identical to Experiment 1, except that the self-paced reading phase was replaced by incidental list-learning and Experiment 2 included additional post-story components: story description and a test of free recall.

**Incidental list-learning.** Participants were presented with a list of 268 words, one at a time. The word list was manually curated to convey the gist of the original Carver story (see[37] for the full list). Each word was onscreen for 1 s before it was replaced with two buttons, which participants used to make a forced-choice decision. The decision depended on the participant's randomly-assigned condition: Italic, Tangible, Theme or Story. The depth with which participants encoded the words in the list was manipulated from shallow (Italic) to deep (Theme/Story), while holding the word list itself constant. Participants in the Italic condition decided whether each word was italicized or not (button labels: Italic or Normal). Sixty-seven of the words were italicized, which was approximately 25% of the total list. For the Tangible condition, participants decided whether or not each word was something concrete that could be touched or inhabited (button labels: Tangible or Intangible), again with 25% of words as targets. For the Theme and Story conditions, participants were informed that the list of words they would see was not random – but instead constructed to have a hidden meaning. In the Theme condition, participants were told that the majority of words (75%) would share a common theme, while a subset of the words (25%) would be unrelated decoys. Specifically, participants were instructed that theme words would "feel like they share something with one another, like they belong in the list". In the Story condition, participants were told that the majority of words (75%) would be ordered in such a way that they could tell a story, while a subset of the words (25%) would be unrelated decoys. Participants were instructed that a story "takes place somewhere", has "characters who have their own thoughts, feelings and emotions", and follows "these characters through a series of situations that affect their lives". In the Theme and Story conditions, participants had to decide whether

each word was a decoy (button labels: Decoy or Theme; Decoy or Story, respectively). To help participants develop a sense of which words belonged to the hidden theme or story, the first 15 words in the words list all belonged to the story or theme. These words were presented in blue lettering, to make them more distinctive. Participants in the Story and Theme conditions were instructed that font colour indicated that a word was not a decoy, while participants in the Italic and Tangible conditions were instructed to make their decisions irrespective of font colour. Decoy words were pseudorandomly interspersed within the list with an average of 6.01 (SD = 3.47) story words between consecutive decoys. Across all conditions, participants received feedback for each decision in the form of a checkmark or an X that appeared above the button they selected for 500 ms, followed by another 500 ms of a blank screen before the onset of the next word. The decision portion of each trial was self-paced.

**Questionnaires, story comprehension and recall.** After post-task free association, participants were informed that the list of words they saw in the list learning task had a hidden meaning – specifically, they were ordered in a way that could convey a story. Participants were given the same definition of a story as the participants who were assigned to the Story condition (for details, see description of Story condition above). Next, participants were asked to generate 10-words that related to the central themes and ideas of the hidden story. Considering many participants who were not in the Story condition may have not noticed a hidden story at all, they were encouraged to guess if they were not sure and were allowed to enter fewer than 10-words if they could not generate that many.

Next, participants completed an edited version of the Narrative Transportation Questionnaire[35], which sought to measure the extent to which they were transported into the hidden story. Then, participants were asked to (i) type a summary of the hidden story in their own words, and then performed (ii) a free recall test, in which they were asked to recall as many of the words from the original word list as possible. During free recall, participants typed words into the center of a blank screen and pressed Enter to submit them. After pressing Enter, the word disappeared. After the free recall, participants the identical multiple choice comprehension test used in Experiment 1 for the Carver story. Finally, participants answered questions about their demographics, the strategies they used, and their subjective experience of lingering, all using an identical format to Experiment 1.

**Word list.** The list of 268 words, as well as the colour and typeface of each word, were identical for all participants in all four incidental learning conditions. The list was designed to convey the gist of the original Carver story, while simultaneously lending itself to the four separate decision tasks (Italic, Tangible, Theme, and or Story). To this end, 201 of the words (~75%) were manually selected from the corpus of participant-generated theme words related to the original Carver story (Experiment 1). The remaining words were highly positively valenced words, selected from[85] and unrelated to the story. The total of 268 words was set to reflect the number of sentences in the original Carver story, such that participants would provide a comparable number of responses across both experiments. For full word list, see[37]. The first 15 words were all related to the Carver story and were presented in the following order: "Claire", "Stuart", "couple", "small-town", "Stuart", "buddies", "camping", "fishing", "find", "girl", "dead", "Claire", "suspicion", "mistrust", "murder". Some example unrelated decoy words are "cheerful", "bliss", "luxury", "peaceful", "happy", "magical". The word list was generated using the following constraints: 25% of words (67 words) had to be italicized, 25% of words had to be tangible (i.e., representing something concrete that could be touched or inhabited), and 25% of words had to be narrative decoys (i.e., unrelated to the story). Only words that were related to the story were selected as targets for the Italic and Tangible conditions to further ensure that

participants in these conditions were attending to story-related words. The target words for the Theme and Story conditions (i.e., unrelated decoys; see description of Incidental list-learning task above), were selected to be both highly unrelated to the story content and positively valenced to help participants learn to discriminate them from story- or theme-related words.

**Participants.** 769 participants took part in Experiment 2 and were recruited via Amazon Mechanical Turk. Data were collected during July 2020. Participants were paid $6.00 USD for their participation and provided informed consent before participating.

After exclusions and quality-assurance checks (see SI: Supplemental Methods), a total of 320 participants were included in the final sample ($N_{male} = 201$; $N_{female} = 113$, with 6 participants selecting "None of the above / Prefer not to identify"). Median age range in the final sample was 35–39 years of age ($Q_2 = 25$–29, $Q_3 = 45$–49, min = 18–19, max = 70–74). Eighty participants were included in each condition: Italic, Tangible, Theme and Story.

**Analysis of free association chains.** Free association data were analyzed using the same document classification and theme similarity analyses as described in Experiment 1. Using the full sample, the most prevalent 10 theme words for this experiment were: "murder", "Claire", "Stuart", "body", "camping", "girl", "death", "friends", "crime", "family". As proper nouns (e.g., "Claire", "Stuart") are unlikely to have the same semantic/distributional properties as the remaining words, they were excluded from the list. Furthermore, as "body" was a cue word in this experiment, it was also excluded. Thus, the final theme words were as follows: "murder", "camping", "girl", "death", "friends", "crime", "family", "investigation", "river", "couple". As in Experiment 1, the 10 theme words used for a given participant's theme similarity calculation were selected after excluding that participant's own theme words.

**Experiment 3: neutral cue variant**
**Experimental procedure.** Experiment 3 was a preregistered follow-up experiment (https://aspredicted.org/qh3dv.pdf; January 3rd, 2022) that sought to test the question of whether or not the lingering observed in Experiments 1 and 2 were a result of our use of a story-related cue to begin free association. Procedures were largely identical to Experiment 1, except that the pre- and post-story free association used a neutral rather than story-related cue: "Enter a word to begin!". In addition to the neutral cue, participants in Experiment 3 performed additional self-report questionnaires, appended to the end of the procedure in Experiment 1. These questionnaires included a more detailed probing of the lingering experience, both in terms of its intentionality and content, and questionnaires examining variables related to mental health. For more details, see SI: Supplemental Methods, Supplementary Note VIII and X. All procedures followed those described in the preregistration protocol.

**Experiment 4: manipulating depth of processing with coherent stories**
**Experimental procedure.** Experiment 4 was a preregistered follow-up experiment (https://aspredicted.org/xd38t.pdf; January 21st, 2022) that sought to determine whether we could limit how deeply we process a coherent narrative, and thereby reducing lingering? Procedures were largely identical to those in Experiment 1, except that the self-paced reading task was modified such that each sentence was presented in the context of one of two cover tasks, manipulating the depth with which participants read an intact story. All participants read a version of the intact Carver story, where half were randomly assigned to a condition that encouraged shallow processing of the story (i.e., proofreading the text for spelling and font errors) and another that encouraged deep processing (i.e., rating the valence of each sentence in the story). For more details, see SI: Supplemental Methods,

Supplementary Note IX and X. All procedures followed those in the preregistration protocol, with the exception of an additional participant exclusion criterion ensuring that participants in the Proofread condition were indeed detecting errors above chance. This criterion was accidentally omitted from the original protocol.

**Statistical assumptions**
In general, parametric analyses (i.e., t-tests, ANOVAs) were applied when the dependent variable was measured at either a ratio or interval scale and when the assumption of independence was met. Nonparametric tests (i.e., Mann-Whitney U tests, Kruskal-Wallis rank sum test, permutation tests) were used in all other situations. Deviations from normality did not influence our choice of whether a parametric or nonparametric test was used. Many parametric tests are robust to nonnormal data, particularly when sample sizes are large[86–88].

**Citation diversity statement**
Recent work in several fields of science has identified a bias in citation practices such that papers from scholars from underrepresented groups are under-cited relative to the number of such papers in the field[89–92]. Here we sought to proactively consider choosing references that reflect the diversity of the field in thought, form of contribution, gender, race, ethnicity, and other factors. First, we obtained the predicted gender of the first and last author of each reference by using databases that store the probability of a first name being carried by a woman[92,93]. By this measure (and excluding self-citations to the first and last authors of our current paper), our references contain 13.43% woman(first)/woman(last), 21.55% man/woman, 14.39% woman/man, and 50.63% man/man. This method is limited in that a) names, pronouns, and social media profiles used to construct the databases may not, in every case, be indicative of gender identity and b) it cannot account for intersex, nonbinary, or transgender people. Second, we obtained the predicted race of the first and last author of each reference by databases that store the probability of a first and last name being carried by an author of color[94,95]. By this measure (and excluding self-citations), our references contain 5.3% author of color (first)/author of color(last), 14.19% White author/author of color, 17.64% author of color/White author, and 62.87% White author/White author. This method is limited in that a) names and Florida Voter Data to make the predictions may not be indicative of racial or ethnic identity, and b) it cannot account for Indigenous and multiracial authors, or those who may face differential biases due to the ambiguous racialization or ethnicization of their names. We look forward to future work that could help us to better understand how to support equitable practices in science.

**Reporting summary**
Further information on research design is available in the Nature Research Reporting Summary linked to this article.

## Data availability
All data and materials from this study are available on Open Science Framework (https://osf.io/dmbx4/)[37]. Data used to generate each figure in the manuscript and supplement are provided in the Source Data file. Source data are provided with this paper.

## Code availability
Analysis code is available on Open Science Framework (https://osf.io/dmbx4/)[37].

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

## Acknowledgements

The authors gratefully acknowledge the support of the Natural Sciences and Engineering Research Council of Canada (postdoctoral fellowship to B.B.), the National Institutes of Mental Health (grant R01MH119099 to C.J.H.) and the Alfred P. Sloan Foundation (research fellowship to C.J.H.). The authors would also like to thank: Tyler Tomita for discussions about machine learning and statistics; Kevin Himberger for advice on online experiments; Heidi Vornbrook Roosa for discussions about narratives and guidance in developing the re-written version of the Carver story; Donna Addis, Tarek Amer, Kristijan Armeni, Simon Brown, Iva Brunec, Janice Chen, Nick Diamond, Katherine Duncan, Hongmi Lee, Yoonjung Lee, Raymond Mar, Vincent Man, Shima Moghaddam, Morris Moscovitch, and Lisa Musz for thoughtful comments on these data and/or manuscript; and Christiane Marie Canillo, William Fisher and Christina Fernando for assistance in organizing the supplemental information and Open Science Framework repository.

## Author contributions

B.B.: Conceptualization (equal); Investigation (lead); Formal analysis (lead); Writing – original draft (lead); Writing – review and editing (equal). A.M.: Writing – review and editing (equal). C.J.H.: Conceptualization (equal); Writing – review and editing (equal); Funding acquisition (lead).

## Competing interests

The authors declare no competing interests.
