## [Peer Review File · Nature Communications]

Narrative thinking lingers in spontaneous thoughtReviewer #1 (Remarks to the Author):

Summary

Bellana, Mahabal, and Honey present an interesting set of behavioral results suggesting that thinking more deeply about words causes their meanings to linger longer in our minds and exert greater influence over our subsequent thoughts. In one set of experiments, participants performed a free association task before and after reading a short story. The post-story associations tended to contain more words that were thematically related to the stories, but this effect diminished somewhat when the story's sentences were scrambled, and diminished even more when the story's words were scrambled. The authors also evaluated "lingering" by examining how this effect persisted over different time intervals within the association task. They found that the story's influence persisted for longer for the intact (vs. scrambled) story conditions. In a second experiment, the authors asked participants to engage free association before and after a list-learning task (instead of reading a story). In different conditions, participants were asked to perform different cover tasks that varied in how much depth of processing they were expected to evoke (e.g. attending to surface features of the words versus attempting to figure out the hidden theme or story reflected by subsets of the words). Deeper processing led to greater self-reported lingering in memory of the studied words. Overall, this is a highly original and well-executed study that adds an important datapoint to the episodic memory and situation understanding literatures. The analyses are appropriate and the results are robust, and the paper seems well-suited to publication in Nature Communications. I have several suggestions for potential improvements:

Major:

1. The authors describe scrambling the story as affecting the "depth of processing." It is certainly plausible that scrambling would affect the deep meaning of the text and/or how someone engages with it. But how could this be explicitly measured or defined? The authors also measure (by self-report) narrative transportation and comprehension. The relations between these measures should be more clearly articulated. For example, are each of these measures getting at essentially the same latent property (e.g., something akin to "engagement")? Or do they differ in important ways? This issue seems central to interpretation of the results, particularly with respect to the potential causal or directional links between depth of processing, transportation, comprehension, and lingering.

2. The authors report a regression-based analysis (e.g., bottom of page 10). However, the details of how this analysis was carried out seem to be missing from the manuscript.

3. The authors' primary measure of "spontaneous thought" was obtained by examining free association responses. However, these responses were not entirely "free," since the free association chains were cued using words that were related to the stories (p. 17, top). The authors make a strong case that the pre-story versus post-story responses differ. But would they have observed the same pattern if the associations had been cued in a much different "region" of semantic space? I don't think a new experiment is necessarily needed here (although that might be the cleanest approach to answering this question). But some discussion of how those cues might have affected or influenced the results would be useful.

Minor:

4. More information is needed regarding how SVMs were applied to classify responses as pre- vs. post story (or word-list). In particular, what were the input features? Were they simply word counts? Or was order accounted for in some way?

5. Also related to the SVM analyses, some additional information is needed with respect

to how the cross validation component was implemented. For example, did the procedure leave out one participant's data at a time? Or were the predictions somehow made within-participant?

6. Are there particular parts of the stories that drive "lingering" and/or "influence"? For example, if you segment the stories into sections or events, can you identify specific sections as displaying an outsized influence on the post-story free associations and/or lingering for especially long?

7. The comments in the discussion section on the "mechanistic neural underpinnings of deep processing" seem out of place in a purely behavioral study. Granted, the processes described in the paper must happen in the brain. But it's not clear what those references add to the current manuscript (e.g., first paragraph on page 15, comments about hippocampal ensembles in the first full paragraph on page 16). I would recommend removing those aspects of the discussion (I feel they distract from the main findings and message). Or alternatively, if the authors feel they are important, some additional unpacking of how and/or why those findings tie into the present study would be useful.

Reviewer #2 (Remarks to the Author):

In this study, Bellana and colleagues implemented a series of studies to examine the extent to which narrative content remains in spontaneous thought for several minutes. In the main studies, participants either read stories with intact sentence structure, order of sentences scrambled or word order scrambled. Before and after these stories, they completed a free association task. Individuals in the intact sentence structure condition are inferred to have engaged in deep processing/encoding of the stories. The authors found that both subjective (as indexed by self-reports) and objective measures (as indexed by document classification and semantic analyses on the free association task) of lingering was strongest in participants who read stories with intact sentence structure. This is especially true in cases where participants reported being "transported" into the stories. The experimental design is clever and well thought-out, and the results are interesting. I appreciate the authors' efforts to implement additional studies and control analyses to address alternative explanations in their main studies, which help strengthen and clarify their findings. Below are some questions that aim to help improve the manuscript.

General Comments

1. Although the manuscript is framed as the influence of deep processing on the subsequent content of spontaneous thought, the experimental conditions are reminiscent of studies examining the impact of depth of processing on memory. The authors have used these memory findings to explain their findings in the discussion. But I encourage a more in-depth discussion of the authors' perspective on how their findings contrast with or extends what is established in memory (e.g. are they tapping into the same cognitive domains merely manifested in different ways, or are they reflecting unique cognitive functions?). This would help highlight the novel contribution of this study.

2. I understand why authors would interpret narrative meaning as a form of deep processing. But I'm less convinced that the word scrambled condition taken to reflect less-deep processing is a fair comparison. At a single word level, this type of comparison makes sense in the context of judging whether the word is italicized versus judging whether the word belongs to a theme. At the story level however, I wonder if the bar for superficial processing would need to be set higher. In comparing superficial versus deep processing, I would think at the very least the content needs to be understood to the same extent (which is not the case given the difference in comprehension performance). So while I understand the value of maximizing conditional differences to elicit the strongest effects, it appears that a more realistic superficial processing condition be one in which participants are asked to focus on the superficial aspects of the story (not necessarily on the physical stimuli itself). For example, a story condition in which

participants' task was to count the number of times a word emerged, versus a condition in which participants' task was to derive the overall meaning of the story. This would allow for even the words/sentence order to be identical across conditions, leaving only the depth of processing to differ. It would seem to me that these two conditions (or variants of them) more aptly reflect what the authors are trying to claim, and the superficial processing condition in this case would also be conceivably more realistic. If newly acquired data with these conditions show the same pattern of results, then the claims that depth of processing impact the lingering of content in spontaneous thought would be more convincing to me.

3. In an era in which studies with small sample size are often criticized for being underpowered, studies with large sample size are faced with a different problem. With a large enough sample, a minute difference can turn out to be significant. This is a concern I have with some of the analyses in this paper. It seems important to rule out the alternative explanation that significant effects are artificially driven by large sample sizes. Was a power analysis done prior to data collection? If not, one suggestion is to determine an estimated effect size, and the sample needed to achieve that effect size. Then randomly subsample from the current sample and implement the same analysis.

Minor Suggestions

1. Given the strong effect of transportation in lingering, it would be informative to partial this out in all the analyses examining the effect of condition to ensure that conditional differences are not attributable to the corresponding differences in transportation across groups.

2. Even in the word scrambled condition, there is some evidence of self-report lingering (~2.5 out of 7). I wonder if this type of lingering may be of a different nature compared to the self-report lingering in the narrative condition. For example, I imagine if I was a participant, I may continue to wonder and think about what actually happened in the story in the word scrambled condition (out of confusion), which seems to be different from the type of lingering that are described by participants in the narrative condition. Given participants provided a description of their mental experience, I wonder if the content of the lingering could potentially be informative and distinguish across the three groups?

3. Deeply processed words linger section - Is ANOVA the most optimal way of comparing across conditions? The data does not seem to be normally distributed.

4. Can authors address the magnitude of the differences observed across the different results in the discussion? For example, while transportation and comprehension scores show more notable group differences (Fig S1/2), theme similarity shows less noticeable pre-post differences even when those differences are significant. The duration of the lingering effect also seems less impactful in the real world given the maximum detectable difference lasts approximately 2.5 minutes in the study, which I understand is a result of the experimental design.

Reviewer #3 (Remarks to the Author):

Bellana and colleagues present a manuscript which describes the properties that determine what types of information prospectively persist to shape cognition. Using both self-report and spontaneous word generation measures, the authors show that information lingers more persistently when it is encoded more deeply, perhaps representing greater transportation into the narrative context. situation level meaning. In brief, the authors found that intact narratives resulted in more lingering thoughts than randomly assorted words, and to a lesser extent randomly assorted sentences. In follow-up studies, the authors showed that this effect generalized to other stories, and could be manipulated by participants' goal orientation using a word list paradigm. While I think, the authors are demonstrating a very interesting phenomenon overall, there were elements of the design, analysis, and interpretation of the findings that left me "hanging" on what the underlying mechanism was. In some ways, the paper felt like it

showed me a phenomenon, but fell short of telling me what cognitive operations make this phenomenon occur. To be clear, I think the authors are onto something quite awesome, but it left me wanting more (because it is so cool!). These comments and other are detailed below.

1. In general, the overall theoretical framework seemed a little diffuse. For example, I couldn't really grasp if this was a study on extending semantic priming or understanding contextual influences on prospective memory. In some ways, I found more utility in the practical real-world examples than the underlying theory they were based on. I think not having a solid understanding of what we learn from asking this specific question, dampened the impact of the findings. Specifically, I am not sure if the authors are targeting semantic similarity, depth of processing, transportation, schemas, or situation models. However, as I write this, I get how difficult it is to operationalize these intersecting constructs.

2. One theoretical framework that especially felt unsupported was the focus on transportation. While my gut feels like this is a good construct to study, I wanted to know why it was selected over other alternative explanations (i.e., generation of situation models; schema consistency). Further, I think the data would be more impactful if I could understand how their findings support one theoretical framework over others.

3. I was a little concerned about individual's goal orientation during the main task in Experiment 1. Picking a cue that was embedded in the story seems like there is a bias to have individuals perform some form of memory retrieval. The reason this is problematic, is that if that arena had a "flavor" of retrieval, the findings wouldn't be very different from prior work done on semantic biases on memory retrieval. One area that could be strengthened to address this point is to find a way to quantify participants' open ended reports, and then get a metric of volitional retrieval to use as a potential moderator or confounding variable.

4. I was a little stuck on how to interpret the findings from the classifier. In study 1, this classifier didn't disambiguate from sentence scrambles and intact story. Then, in the follow-up, the classifier was efficacious for some but not all stories. Then in Experiment 2, which I found the most surprising, was that the classifier was unable to disambiguate between conditions. The authors then followed up with a theme similarity analysis which showed a difference for the story condition, but also the theme condition. This pattern of results made it hard for me to interpret what unified interpretation was driving the lingering effects. I also wanted more consideration on what factors may lead to explicit versus implicit lingering, and how this differed across studies.

5. Did the authors collect data on transportation in the word list study. This seemed like a really promising construct, and I am curious how it would differ across the story and them conditions in Experiment 2.

6. The authors state that there is little empirical evidence to support our intuitive sense that 'meaningful' experiences resonate with us, but this precludes a fairly large literature on intrusive memories that could be incorporated into the manuscript.

7. The authors discuss the idea of levels of processing as an organizational principle, which I thought sounded like a promising framework. Given this, additional experiments that tested predictions made from the lens of levels of processing (i.e., transfer appropriate processing) could help strengthen the findings.

8. Did the authors consider doing some sort of cross story, theme analysis? If themes are the major focus of the lingering effect, perhaps a classifier could be fit to disambiguate free associations drawn from different stories, with the prediction that there would be more errors in the two re-writes of the same story versus across story comparisons.

**Sincerely,
Vishnu "Deepu" Murty
Temple University**

Response to Reviewers:

Reviewer comment are in black

Our responses are in blue

Quotes/excerpts from manuscript are in green, with revisions in **bold**

Reviewer #1 (Remarks to the Author):

Summary

Bellana, Mahabal, and Honey present an interesting set of behavioral results suggesting that thinking more deeply about words causes their meanings to linger longer in our minds and exert greater influence over our subsequent thoughts. In one set of experiments, participants performed a free association task before and after reading a short story. The post-story associations tended to contain more words that were thematically related to the stories, but this effect diminished somewhat when the story's sentences were scrambled, and diminished even more when the story's words were scrambled. The authors also evaluated "lingering" by examining how this effect persisted over different time intervals within the association task. They found that the story's influence persisted for longer for the intact (vs. scrambled) story conditions. In a second experiment, the authors asked participants to engage free association before and after a list-learning task (instead of reading a story). In different conditions, participants were asked to perform different cover tasks that varied in how much depth of processing they were expected to evoke (e.g. attending to surface features of the words versus attempting to figure out the hidden theme or story reflected by subsets of the words). Deeper processing led to greater self-reported lingering in memory of the studied words. Overall, this is a highly original and well-executed study that adds an important datapoint to the episodic memory and situation understanding literatures. The analyses are appropriate and the results are robust, and the paper seems well-suited to publication in Nature Communications. I have several suggestions for potential improvements:

We thank the reviewer for their positive assessment of our work and for their thoughtful suggestions. We have extensively revised the manuscript text and have added two new experiments and multiple further analyses. We hope the reviewers will agree that the manuscript has been substantively improved and that we have satisfied their concerns.

Major:

1. The authors describe scrambling the story as affecting the "depth of processing." It is certainly plausible that scrambling would affect the deep meaning of the text and/or how someone engages with it. But how could this be explicitly measured or defined? The authors also measure (by self-report) narrative transportation and comprehension. The relations between these measures should be more clearly articulated. For example, are each of these measures getting at essentially the same latent property (e.g., something akin to "engagement")? Or do they differ in important ways? This issue seems central to interpretation

of the results, particularly with respect to the potential causal or directional links between depth of processing, transportation, comprehension, and lingering.

We thank the reviewer for bringing up these points. Below, we expand on how our measures are related to one another.

Our position is that the extent to which an input (e.g., a passage of text) ‘lingers’ in our spontaneous thoughts can be predicted by how deeply it was processed. Here, depth reflects the amount of ‘situation-level meaning’ that was extracted from the input.

Attending to a word in the context of an incoherent sentence (e.g., in our word-scrambled condition) affords the extraction of less meaning (i.e., less deep processing) than attending to that same word in the context of a coherent sentence in a coherent narrative.

Let us consider the word “four”. On its own, or in the context of unrelated words, it likely evokes an obvious association: a number. However, after reading Miranda July’s *Roy Spivey*, “four” takes on an entirely different meaning: it was the missing digit from a phone number that a famous actor shared with the narrator after flirting on an airplane; it was a number that he asked her to commit to memory in an effort to reify their brief but profound connection; a number he asked her to call when she got home; a secret code; a potential affair; an alternate life; a decision to be made.

Narratives function as a way to embed words into broader situations, thus making the word more ‘meaningful’ (i.e., deep processing). We operationalized our participant-level index of “deep processing” as self-reported narrative transportation. In order to be transported into the ‘world’ of a story, the reader must be able to activate the relevant information to build said world (e.g., a situation model, Buselle & Bilandzic, 2008). Therefore, the extent to which a person reports feeling immersed in a story can be used as a proxy of the number of meaningful associations that passage of text was able to evoke (i.e., depth).

Our model would be:

1. Deep processing = extracting situation-level meaning (as opposed to low-level lexical semantics or perceptual features of an input)
2. Narrative transportation = experience of deeply processing a narrative
3. Deep processing → Lingering

We take the fact that our scrambling procedure reduced transportation as evidence that scrambling limited how deeply the text was processed. To lend additional support to our claim that ‘deep processing’ drives lingering, we conducted an additional experiment (SI: Supplemental Results IX - *Shallow processing of a coherent story reduces lingering [Experiment 4]*). We elaborate on this experiment in response to Reviewer 2’s second

point (see below). In brief, we directly manipulated the depth with which participants processed coherent texts (i.e., judging the emotional quality of each sentence vs. proofreading each sentence for spelling or font errors) and report more evidence of lingering for the deeper condition (i.e., emotion). These results converge with those from our scrambling procedure and those from the word-list experiment (Experiment 2), and further emphasize that deep processing indeed drives lingering.

Furthermore, a detailed comprehension test could also provide a good operational index of depth of processing. The extent to which it measures inferential knowledge and deep understanding of the relationships between entities in the text should reflect the extent to which situation-level meaning was extracted. Unfortunately, the specific comprehension test that we employed in our study was not of this type. We operationalized story comprehension using a 24-item multiple choice test, with some questions tapping into the plot (e.g., “What did Stuart do after calling the police?”) and others tapping into more general familiarity with the text itself and not its meaning (e.g., “Which of the following beverages figured most prominently in this passage?”). Furthermore, readers who find they are able to make sense of the story, but do not interpret it in the way that was intended by the author, will not necessarily perform well on a comprehension test. We prefer transportation as a subjective index of deep processing for these reasons.

Like the reviewer suggests, deep processing and transportation do relate to the idea of ‘engagement’, however, the point we hope to convey is that they entail a specific *kind* of engagement, emphasizing accessing and representing situation-level meaning. For example, you could watch a movie and engage with the relatively shallow-level elements of the input (e.g., the colours) rather than the deeper levels (e.g., plot, emotions, situations, consequences). We had originally described this point in the introduction, with our example with the ant (“**However, there is also an endogenous component ...**”; page 3).

We have revised the manuscript in several places to better communicate the points described above. See example excerpts below (revised text in bold):

Introduction (page 4): “We found that when participants read coherent narratives, the themes **and details** of the story lingered for several minutes in their subsequent free association chains, more so than in participants who read scrambled versions of the same text. This observation was replicated across multiple stories. **In a follow-up experiment, we additionally demonstrate that coherent stories linger more when participants judge the emotional properties of each sentence rather than their spelling or font. Finally,** we also observed the lingering effect **after** participants narrativized a list of words, but not when they judged their perceptual properties (e.g., italic type). Overall, regardless of the objective coherence of the text, participants’ subsequent experience of lingering was **best** predicted by **the degree to which** they felt transported by the material while reading.

We propose that a “depth of processing” framework provides a compelling account for our results. According to this framework, engaging with a text’s typography is shallower than engaging with word-level semantics²⁴ and the extraction of situation-level meaning from a text is deeper than word- or even sentence-level semantics²⁷. When applied to our data, a clear pattern emerges: deeper processing predicts greater subsequent lingering. More generally, our data indicate that more elaborative styles of thinking, such as the construction of situation models while reading stories, produce an especially long-lasting mental context. ”

Results: Scrambling limits deep processing (page 4): “Transportation was measured using a 13-item modified version of the Narrative Transportation scale³³. Transportation requires participants to attend to deeper, narrative-level meaning³⁴ as opposed to word-level semantics. Some example questions include: “While I was reading the text, I could easily picture the events in it taking place”, “I found myself thinking of ways the text could have turned out differently” and “I was mentally involved in the text while reading it” (see **SI: Supplemental Methods**). **Thus, transportation reflects the act of building, representing and engaging with a situation model, which we take as a self-reported index of deeply processing a narrative.**”

Results: Transporting stories linger (page 11): “Why do coherent stories linger in our minds? **One possibility is that stimulus-level differences (e.g., objective coherence of the text) determine depth of processing, which in turn predicts lingering. Specifically,** coherent stories may contain types of information that are absent **from** scrambled sequences of words or sentences (e.g., agents, actions, intentions, embedded in broader situations evolving over time), **and engaging with this kind of deep, situation-level meaning increases the likelihood of these thoughts persisting in mind. Alternatively, lingering may be better explained by an individual’s appraisal of a stimulus, rather than the properties of the stimulus itself. According to this hypothesis, if an individual is able to extract, represent and immerse themselves in the ‘world’ of a story, it should linger in mind, regardless of the text’s objective coherence. In order to adjudicate between these two possibilities,** we examined whether participant-level narrative transportation ratings, **a proxy for the depth with which an individual engaged with the text,** could predict subjective and objective measures of lingering, independently of whether **the text was intact or scrambled. To this end,** we included **transportation** in a 10-fold cross-validated regression model with backwards stepwise feature selection (see Methods). Regression models included participant-level measures of: transportation by the story; **performance on a comprehension test (i.e., memory for specific verifiable details from the story);** and experimental condition (i.e., Intact / Sentence-scrambled / Word-

scrambled). **If lingering is determined by properties of the text itself, experimental condition should be the best predictor. If lingering instead results from the depth with which an individual engages with the text, irrespective of objective coherence, transportation should outperform all other predictors.**"

Discussion (page 15): "Critically, these lingering influences were strongest amongst participants who reported being 'transported' into the story world, **arguably reflective of deep processing**³⁰, regardless of the objective coherence of the text they read (Figure 5)."

2. The authors report a regression-based analysis (e.g., bottom of page 10). However, the details of how this analysis was carried out seem to be missing from the manuscript.

We thank the reviewer for catching this. Additional details regarding our model implementation have been added to the Methods section of the manuscript.

Methods: Analysis of free association chains (Page 21-22): "***Predicting lingering via stepwise feature selection. To determine the measures that best predicted our subjective (self-reported lingering) and objective (theme similarity) measures of lingering, we performed stepwise feature selection. Predictors input into the regression model were: participant-level scores on transportation***³³, participant-level performance on the 24-item multiple choice comprehension test, and experimental condition (i.e., Intact / Sentence-scrambled / Word-scrambled). Self-reported lingering and theme similarity ($M_{post-story} - M_{pre-story}$) were included as dependent variables, in two separate regression models. Stepwise feature selection was performed using a backwards stepwise linear regression with 10-fold cross-validation, implemented in R using caret (model = leapBackward)⁷⁹. All numerical measures were z-scored prior to inclusion in the model."

3. The authors' primary measure of "spontaneous thought" was obtained by examining free association responses. However, these responses were not entirely "free," since the free association chains were cued using words that were related to the stories (p. 17, top). The authors make a strong case that the pre-story versus post-story responses differ. But would they have observed the same pattern if the associations had been cued in a much different "region" of semantic space? I don't think a new experiment is necessarily needed here (although that might be the cleanest approach to answering this question). But some discussion of how those cues might have affected or influenced the results would be useful.

We thank the reviewer (and Reviewer 3) for bringing up this point. We agree. In fact, we opted to use story-relevant cue words when designing our experiment precisely because

we thought it should help amplify any lingering in free association. But it remains unclear if lingering occurs *because* of our use of a story-related cue, or if it would occur irrespective of the cue word.

To this end, we conducted an additional preregistered experiment (Experiment 3; <https://aspredicted.org/qh3dv.pdf>) in which we had a new set of 80 participants read the intact version of the Carver story. In this new version of the experiment, we dropped the story-related cue words from pre- and post-story free association. Instead, at the beginning of free association, participants saw the following instruction: “Enter a word to begin!”

As predicted, we found evidence for lingering in our subjective (self-report) and objective measures (document classification and theme similarity in free association). Of note, the theme similarity analyses revealed that the effect was strongest immediately after the story ended and returned to baseline after the 20th free associates – suggesting that the timecourses originally reported in the manuscript (Figure 4B) were not a result of the story-related cue word.

The details of this experiment are briefly mentioned in the manuscript [Methods: Experiment 1 (page 19)], but reported in full in the Supplemental Information (SI: Supplemental Results VIII). A summary figure is included below for convenience.

Figure S10. Linger is not a result of using story-related cue words in free association (Experiment 3). (A) Histograms revealed that participants continued to self-report the experience of lingering despite being presented with a neutral (“Enter a word to begin!”) rather than a story-related cue during free association. Participants provided their rating on a 7-pt scale: 7 = very much, 1 = not at all. Black dashed line represents the mean rating per condition. (B) Results of document classification again indicate that pre- and post-story free association chains were discriminable. Classifiers were trained on all subjects from this experiment ($n=80$), using a leave-one-subject-out cross-validation procedure with 500 bootstraps. Dashed line represents the mean classification accuracy. Null distributions are plotted in gray. (C) Theme similarity pre- and post-task highlights evidence for an increase in similarity to story themes after reading. Grey lines show the change in theme similarity within-subject. Group means are displayed using black circles. Error bars represent 95% confidence intervals. For display purposes, significance was estimated with paired-sample t-tests comparing pre- vs. post-task theme similarity [ns $p > .05$; * $p < .05$; ** $p < .01$, *** $p < .001$, **** $p < .0001$]. (D) Timecourse of post-story theme similarity displayed using 10-associate windows. Despite using a cue that was not directly related to the story, theme similarity was strongest immediately after the story ended. Effect size was calculated using Cohen’s d , comparing theme similarity post-story minus pre-story within each window. Error bars represent 95% confidence intervals.

Minor:

4. More information is needed regarding how SVMs were applied to classify responses as pre- vs. post story (or word-list). In particular, what were the input features? Were they simply word counts? Or was order accounted for in some way?

We have added additional details to our original description of our SVM-based document classification analysis. Please see the relevant excerpt from the manuscript below. Again, the bolded text was newly added to the revised manuscript.

As described below, the input features were the word counts (mean-centered within-participant, and then scaled across-subjects). The model did not make use of word order.

Methods: Analysis of free association chains (Page 20): “*Document classification*. To determine whether pre- vs. post-story free association chains were statistically discriminable, we used a support vector machine (SVM, implemented in R⁷⁸) to perform document classification. A document-term matrix was computed from all free association chains belonging to participants from a given condition, for a given story. Each row of the matrix represented a free association chain, with columns for every unique word from all free association chains, and the matrix values composed of each word’s raw count in each chain. Each participant is associated with two rows in the document-term matrix: one for pre-story free association and another for post. Word counts were rescaled by mean centering each column (i.e., word) within-participant, and then dividing the full column by the across-participant standard deviation. Using a leave-one-participant-out cross validation procedure, we trained a linear SVM to discriminate between pre- and post-story free association chains in the document-term matrix. **Specifically, the input features to the model were the term-frequencies for a held-out participant’s free association chain (i.e., a held-out row in from the document term matrix). The model output was a prediction as to whether the input was generated pre- or post-story.** We repeated this analysis 500 times and report the mean as our estimate of true classification accuracy. **Note that the model’s predictions are based on whether or not a word was in a word chain and does not account for the order words appeared in free association.**”

5. Also related to the SVM analyses, some additional information is needed with respect to how the cross validation component was implemented. For example, did the procedure leave out one participant's data at a time? Or were the predictions somehow made within-participant?

As described in the original Methods section of the manuscript, we used “a leave-one-participant-out cross validation procedure” (see above excerpt). In other words, within a given condition, the model was trained on the pre- and post-story free association chains

for all subjects, except one held-out subject ($n = 79$). Then, the trained model was used to independently assign a binary label (i.e., “pre-story” or “post-story”) to both the pre- and post-story free association chains from the held-out participant. Accuracy was then computed by comparing the assigned binary labels against ground truth, and we reported accuracy averaged across all folds of the cross-validation procedure.

6. Are there particular parts of the stories that drive "lingering" and/or "influence"? For example, if you segment the stories into sections or events, can you identify specific sections as displaying an outsized influence on the post-story free associations and/or lingering for especially long?

This is an excellent question. However, the stories for which we have data are mostly composed of recurrent themes, which makes this possibility difficult to test. For example, in the Carver story, there were multiple sections relating to “murder” and “death”, which were participant-generated theme words from the story; in the July story, many sections were related to the idea of an “affair” or “secret”, also theme words from the story. Also, this kind of recurrence is common of stories in general. It’s not often that a story is composed of thematically discrete sections.

An interesting possibility might be to have participants read two thematically-separable stories back to back and examine lingering afterwards. While interesting, we believe this to be outside the scope of the current manuscript.

7. The comments in the discussion section on the "mechanistic neural underpinnings of deep processing" seem out of place in a purely behavioral study. Granted, the processes described in the paper must happen in the brain. But it's not clear what those references add to the current manuscript (e.g., first paragraph on page 15, comments about hippocampal ensembles in the first full paragraph on page 16). I would recommend removing those aspects of the discussion (I feel they distract from the main findings and message). Or alternatively, if the authors feel they are important, some additional unpacking of how and/or why those findings tie into the present study would be useful.

We thank the reviewer for this point. Nonetheless, we included this section because we believe it provides grounds for a “rational” explanation regarding *why* lingering might occur. We state this point explicitly in the first two sentences of our final paragraph:

Discussion (page 17): “Having considered which kinds of processing increase mental lingering, we must finally ask: ‘why linger at all’? In an ever-changing world, why should any of our experiences colour the trajectory of our thoughts for minutes after they end?”

Since “replay” of our recent experiences appears to be a fundamental building block of how we consolidate our memories (e.g., Foster, 2017; Tambini & Davachi, 2019), we

included this section to highlight the possibility that experiences that linger may be prioritized in later memory. We mentioned this point in the discussion: “experiences associated with deep thinking may be preferentially replayed, in turn helping us prioritize significant or meaningful events in memory^{70,71}” (Discussion; page 17).

We believe this point concerns *why lingering might benefit us*, which falls within the scope of a behavioural manuscript, despite some of the background coming from neuroscience.

Reviewer #2 (Remarks to the Author):

In this study, Bellana and colleagues implemented a series of studies to examine the extent to which narrative content remains in spontaneous thought for several minutes. In the main studies, participants either read stories with intact sentence structure, order of sentences scrambled or word order scrambled. Before and after these stories, they completed a free association task. Individuals in the intact sentence structure condition are inferred to have engaged in deep processing/encoding of the stories. The authors found that both subjective (as indexed by self-reports) and objective measures (as indexed by document classification and semantic analyses on the free association task) of lingering was strongest in participants who read stories with intact sentence structure. This is especially true in cases where participants reported being “transported” into the stories. The experimental design is clever and well thought-out, and the results are interesting. I appreciate the authors’ efforts to implement additional studies and control analyses to address alternative explanations in their main studies, which help strengthen and clarify their findings. Below are some questions that aim to help improve the manuscript.

We thank the reviewer for their positive assessment of our work, particularly our attempts to be thorough with respect to alternative accounts. We have addressed all of their concerns below and hope that they agree with us that the manuscript is considerably improved.

General Comments

1. Although the manuscript is framed as the influence of deep processing on the subsequent content of spontaneous thought, the experimental conditions are reminiscent of studies examining the impact of depth of processing on memory. The authors have used these memory findings to explain their findings in the discussion. But I encourage a more in-depth discussion of the authors’ perspective on how their findings contrast with or extends what is established in memory (e.g. are they tapping into the same cognitive domains merely manifested in different ways, or are they reflecting unique cognitive functions?). This would help highlight the novel contribution of this study.

We thank the reviewer for bringing up this point. We agree that the relationship between lingering and existing accounts of memory warrants further consideration.

One important distinction between our experiment and more traditional examinations of “deep processing” (e.g., Craik & Lockhart, 1972) comes down to our outcome measures. Typically, levels of processing paradigms use explicit, goal-directed memory tests, such as free recall or recognition. In our paradigm, our test is not one of memory but instead a measure of spontaneous thought. In the original manuscript, we tried to highlight this distinction by emphasizing the unbidden and unintentional quality particular to lingering.

Results: Stories elicit a lasting influence on spontaneous thought (page 5): “In open-ended descriptions of their experience, participants often described lingering with an unintentional quality, distinguishing it from volitional rehearsal (e.g., “*In the first round, the words I typed were considerably more organic than those in the second round, as I could not really get the story out of my head after reading it, so many of the associations were related to extraneous thoughts or associations with the story itself*”).”

To further examine this possibility, we included a more detailed probing of participants’ self-reported lingering in two additional experiments, conducted to address other points brought up by yourself and the other reviewers. Specifically, participants who indicated that they experienced some lingering (i.e., reporting a score of 2 or more on our 7-pt self-reported lingering scale; Question: “To what extent did the text linger in your mind after reading it?” ; 1 = “Not at all”, 7 = “Very much”) were asked additional questions about their experience.

First, participants were asked the following question: “Were you *intentionally* reflecting on the text while playing the word chain game? Or, did aspects of text come to mind *unintentionally*?” Participants had to choose one of five potential responses: (i) “I was intentionally reflecting on the text I read”, (ii) “The text I read came to mind unintentionally”, (iii) “Both”, (iv) “Neither”, (v) “Don’t know”. The following waffle plots display the results for Experiment 3 (Neutral cue) and Experiment 4 (Proofread vs. Emotion). Each cell represents a participant (n = 80 per condition, excluding the participants who reported a 1, i.e. “Not at all”, on the self-reported lingering scale).

Figure S15. *Lingering may be outside of our volitional control.* Participants in Experiment 3 and 4 answered additional questions about the volitional quality of their experience of a story lingering in mind, if they experienced any lingering at all. Most participants endorsed lingering as unintentional, distinguishing it from rehearsal and the kinds of explicit memory that traditionally benefit from deep processing (Craik & Lockhart, 1972). Plots were generated in R using `waffle()` (Rudis, 2017).

Indeed, the majority of participants reported that lingering was unintentional. In the open-ended descriptions, several participants described trying to deliberately counteract lingering:

“I think I almost tried to not use words/themes that were in the text as I didn’t want to have been influenced by the text. I realised I was coming up with blanks/dead ends a little more the second time as I didn’t want to go towards darker themes or water based themes”

Here, we make the point that deep processing not only improves performance on traditional explicit memory tests but emphasizes a lasting ‘mental context’ that can shape the content of our spontaneous thoughts, even in the absence of volition.

We have made this point explicit in the results and discussion of the revised manuscript (see below), alongside more details in the supplement (SI: Supplemental Results XI – *Is lingering volitional?* [Experiments 3 & 4]). Note: bolded text reflects changes in the revised manuscript.

Results: Stories elicit a lasting influence on spontaneous thought (page 5):
“**Lingering was usually involuntary.** In open-ended descriptions of their experience, participants often described lingering with an unintentional quality, distinguishing it from volitional rehearsal (e.g., “*In the first round, the words I typed were considerably more organic than those in the second round, as I could not really get the story out of my head after reading it, so many of the associations were related to extraneous thoughts or associations with the story itself*”). **In a separate sample, we directly asked participants to describe the volitional quality of lingering: 51% described it as unintentional, only 7% as intentional, 18% as both, with the remaining 24% of participants describing it as neither or unsure (see *SI: Supplemental Results XI*; for all open-ended descriptions, see *SI: Supplemental Materials*).**”

Discussion (page 15): “Although participants’ experiences of lingering varied, many described the ongoing mental presence of the text as something unbidden or even distracting: “*I think maybe the story stayed with me and affected me a little. I tried to not let it influence me and to go where the words took me*”. **In fact, when we asked a separate group of participants to describe the volitional quality of their lingering experience, the majority of them confirmed that it was unintentional (see *SI: Supplemental Results XI*).** Thus, the lingering phenomenon **appears to be distinct from intentional rehearsal or the kinds of explicit memory that typically benefit from deep processing²⁴, but instead acts as a kind of latent constraint^{5,7} on participants’ spontaneous thought.**”

2. I understand why authors would interpret narrative meaning as a form of deep processing. But I’m less convinced that the word scrambled condition taken to reflect less-deep processing is a fair comparison. At a single word level, this type of comparison makes sense in the context of judging whether the word is italicized versus judging whether the word belongs to a theme. At the story level however, I wonder if the bar for superficial processing would need to be set higher. In comparing superficial versus deep processing, I would think at the very least the content needs to be understood to the same extent (which is not the case given the difference in comprehension performance). So while I understand the value of maximizing conditional differences to elicit the strongest effects, it appears that a more realistic superficial processing condition be one in which participants are asked to focus on the superficial aspects of the story (not necessarily on the physical stimuli itself). *For example, a story condition in which participants’ task was to count the number of times a word emerged, versus a condition in which participants’ task was to derive the overall meaning of the story. This would allow for even the words/sentence order to be identical across conditions, leaving only the depth of processing to differ. It would seem to me that these two conditions (or variants of them) more aptly reflect what the authors are trying to claim, and the superficial processing condition in this case would also be conceivably more realistic. If newly acquired data with these conditions show the same*

pattern of results, then the claims that depth of processing impact the lingering of content in spontaneous thought would be more convincing to me.

We generally agree with the reviewer regarding this point. To address this concern, we conducted a new preregistered experiment (Experiment 4; <https://aspredicted.org/xd38t.pdf>) in which participants read an identical, coherent version of the story, while performing different cover tasks that manipulated depth of processing (SI: Supplemental Results IX – *Shallow processing of a coherent story reduces lingering [Experiment 4]*).

160 online participants performed our self-paced reading and free association paradigm. All participants were presented with a version of the original Carver story, without scrambling. Instead, the text was edited such that 50% of the sentences contained an “error”. Errors were either (i) in spelling (e.g., “teh” instead of “the”; or “dihses” instead of “dishes”) or (ii) in font (i.e., most of the story was displayed in a sans-serif font, Arial, while the “error” words would be displayed in a serif font, Times). To ensure overall readability, only 10% of the sentences contained spelling errors.

Participants were randomly assigned to one of two conditions, one that encouraged shallow processing of the story (i.e., Proofread) and another that encouraged deep processing (i.e., Emotion). In the Proofread condition, participants were instructed to read each sentence and indicate, through key press, the total number of errors they found (i.e. response options, 0, 1, or 2). Total number of errors was used such that participants could not only look at the font, but had to also read words to ensure there weren’t spelling errors. Like in Experiment 2, participants received feedback after each trial in the form of a checkmark or x. In the Emotion condition, participants were instructed to indicate how each sentence contributes to the valence of that moment in the story (i.e., response options: positive, negative, neutral). These participants were explicitly instructed to expect and not pay attention to the spelling and font errors in the story. No feedback was provided. For additional details re: procedure, see SI: Supplemental Methods.

In line with our hypothesis that deep processing results in lingering, participants in the Emotion condition reported more lingering than participants in the Proofread condition (Panel A; Mann-Whitney $U = 3882$, $p = 0.018$, $r_{\text{ranked-biserial}} = 0.21$). Self-report converged with our objective measures in that our SVM-based document classifier was able to accurately classify free association chains as pre- or post-story for participants in the Emotion condition but not the Proofread condition (Panel B). Interestingly, theme similarity did not show a difference across conditions when averaging over the entire 5-minutes of free association (Panel C), but a windowed analysis demonstrated that theme similarity was higher for the Emotion condition, specifically in the first 10-words immediately after the story ended (Panel D).

Figure S11. Shallow processing of a coherent story reduces lingering (Experiment 4). (A) Histograms revealed that participants assigned to a cover task emphasizing deep processing of the story (Emotion) reported more lingering than those assigned to a shallow reading condition (Proofread). Participants provided their rating on a 7-pt scale: 7 = very much, 1 = not at all. Black dashed line represents the mean rating per condition. **(B)** Results of document classification again indicate that pre- and post-story free association chains were discriminable for participants assigned to the Emotion but not Proofread condition. Classifiers were trained on all subjects from a given condition ($n=80$), using a leave-one-subject-out cross-validation procedure with 500 bootstraps. Dashed line represents the mean classification accuracy. Null distributions are plotted in gray. **(C)** Theme similarity pre- and post-story fails to show any increase in the semantic closeness to the story themes after reading, for either condition. Grey lines show the change in theme similarity within-subject. Group means are displayed using black circles. Error bars represent 95% confidence intervals. For display purposes, significance was estimated with paired-sample t -tests comparing pre- vs. post-task theme similarity [ns $p > .05$; * $p < .05$; ** $p < .01$, *** $p < .001$, **** $p < .0001$]. **(D)** While theme similarity averaged over all 5-minutes of free association did not show evidence of lingering, examining the timecourses of post-story theme similarity using 10-associate windows revealed evidence of increased theme similarity immediately after reading, particularly with participants in the Emotion condition. Effect size was calculated using Cohen's d , comparing theme similarity post-story minus pre-story within each window. Error bars represent 95% confidence intervals.

These data provide more convergent evidence that deep processing, this time manipulated via cover task performed while reading a coherent story, results in a lasting influence on spontaneous thought.

The relatively small difference between the self-reported lingering in the Emotion and Proofread conditions may be a consequence of cover tasks reducing processing depth relative to free, unconstrained reading of a coherent story. In line with this hypothesis, participants who were permitted to freely read an intact version of the Carver story [i.e., Experiment 1: Carver (Original), Carver (Replication), Carver (Rewrite); Experiment 3: Carver (Original w/ Neutral cue)] reported more lingering than participants in the Emotion condition (Median_{Intact} = 5, Median_{Emotion} = 4.5; Mann-Whitney U test: $U = 10208$, $p = 0.004$, $r_b = 0.20$). Similarly, participants in the Intact condition reported feeling more transported into the story while reading as compared to those in the Emotion condition ($M_{Intact} = 0.63$, $M_{Emotion} = 0.60$; Welch two-sample t-test: $t(124.33) = 2.43$, $p = 0.003$, $d = 0.30$).

Additional details regarding Experiment 4 can be found in the supplement (SI: *Supplemental Results IX*), which are pointed to in the main text as well.

Results: Deeply processed words linger (page 12): “Encoding a given experience can entail attending to its surface-level features (e.g., the verbatim text in a written story) to the broader meaning it represents (e.g., constructing a situation model from the written text^{27,34}), and the more likely we are to attend to its deeper meaning, the more likely it will linger in our minds (for related ideas, see²⁴). **In fact, when participants performed a cover task that encouraged shallow processing, lingering was reduced even when participants read an objectively coherent version of the Carver story (see SI: Supplemental Results IX – Experiment 4).**”

3. In an era in which studies with small sample size are often criticized for being underpowered, studies with large sample size are faced with a different problem. With a large enough sample, a minute difference can turn out to be significant. This is a concern I have with some of the analyses in this paper. It seems important to rule out the alternative explanation that significant effects are artificially driven by large sample sizes. Was a power analysis done prior to data collection? If not, one suggestion is to determine an estimated effect size, and the sample needed to achieve that effect size. Then randomly subsample from the current sample and implement the same analysis.

We thank the reviewer for bringing up the concern of large sample sizes amplifying the importance of small, inconsequential effects. We did not conduct an a priori power analysis before running our experiments, given the novelty of our dependent measure and the lack of previous empirical work on our psychological phenomenon (“lingering”) of

interest. However, we argue that the concern of an excessively large sample size does not apply in our case for a couple reasons.

- (1) **We report effect sizes:** It is true that a tiny within-subjects effect can be statistically significant with a large enough sample size. This is unlikely with respect to self-reported lingering, given the clearly visible shift in the underlying distributions across conditions (e.g., Figure 1C, Figure 2A, Figure 6A). But, it may be relevant to our theme similarity analyses, as the Reviewer states in Minor Suggestion #4: "...theme similarity shows less noticeable pre-post differences even when those differences are significant". Importantly, however, we explicitly report the effect sizes for these results, so the reader can judge their magnitude. The post-story increase in semantic similarity to story themes was twice as large in the Intact condition ($d = 0.56$) as compared to the Sentence-scrambled condition ($d = 0.28$) (see Results: Story themes are upregulated in post-story free association; page 11). Given that effect sizes are robust to differences in sample size (e.g., Cohen's d is calculated by scaling the mean difference by the average standard deviation), our readers can see precisely the size of our effects in a way that is robust to differences in sample size.
- (2) **We established a negative control for our measurement tool:** To put this concretely, our Word-scrambled condition, with an identical sample size ($n=80$) to our other conditions, produced no discernible lingering in free association (i.e., document classification: 52% accuracy, $p = 0.34$; or theme similarity: $t(79) = -0.08$, $p = 0.93$, $d = -0.01$). Similarly, in the Sentence-scrambled conditions, document classification was above chance for only 2/4 of the datasets from Experiments 1, as compared to 4/4 above chance classification for the Intact condition (6/6 if we count newly added Experiment 3 and the deep processing condition from Experiment 4). In other words, our dependent variables can produce a null result.

In spite of the two reasons noted above, we agree with the Reviewer that it can always help to concretize the magnitude of the changes in free association from pre- to post-story. To this end, we have included example odd ratios in the revised manuscript and included histograms displaying the top 20 free association words in terms of post-story and pre-story odds, for all conditions in all experiments, in the Supplemental Information (SI: Supplemental Results XIII – *To what extent does lingering change free association? [Experiments 1, 2, 3 & 4]*). Odds ratios effectively and concretely communicate how likely a word was to occur in post- vs. pre-story free association (e.g., "6 times more likely to occur in post- vs. pre-story free association). However, since odds ratios can be numerically imprecise when estimated from sparse datasets like free association (because of the inclusion of small probabilities in the denominator), we only report examples in the main manuscript and reserve the more complete treatment of these data for the supplement.

Results: Story themes are upregulated in post-story free association (page 9-10): “Both general themes and specific episodic content lingered substantially. Interpreting changes in word frequency as odds ratios [i.e., $p(\text{Post})/p(\text{Pre})$], we observed large odds ratios for both general theme words and detailed content. For example, general themes such as “murder” (odds ratio = 6.3) and “loss” (odds ratio = 12) emerged in the Carver-Original and July stories, respectively. As examples of detailed content, “funeral” exhibited an odds ratio of 12 for Carver-Original, while “four” (odds ratio = 6.5) and “spy” (odds ratio = 6.5) emerged in the July story. Concretely, this means that 1 in 6 participants generated the word “funeral” after reading the intact version of the Carver story, while only 1 in 80 generated the word before the story. Similarly, more than 1 in 5 participants generated the word “four” after the intact version of the July story, compared to 1 in 40 who did so before. Thus, the lingering material includes both general themes and more specific episodic content, and the lingering is strong enough to be practically detectable in a group of people all exposed to a common narrative. For more examples of odds ratios, see *SI: Supplemental Results XIII.*”

Minor Suggestions

1. Given the strong effect of transportation in lingering, it would be informative to partial this out in all the analyses examining the effect of condition to ensure that conditional differences are not attributable to the corresponding differences in transportation across groups.

We appreciate the reviewer’s point here, however, we believe that it is addressed in our feature selection analysis (*Results: Transporting stories linger*). Specifically, when including experimental condition, transportation and comprehension in a 10-fold cross-validated regression model with backwards stepwise feature selection, we find that transportation best predicts subjective (self-report) and objective (theme similarity) measures of lingering. In other words, adding experimental condition to the model did not account for significantly more variance.

Therefore, it is the extent that an individual feels immersed in a story that matters, rather than the objective coherence of the text per se.

We were explicit about this conclusion in the original manuscript.

Results: Transporting stories linger (page 12): “Thus, the extent to which an individual felt transported into the story world was an important determinant of post-story lingering, explaining more out-of-sample variance than their experimental condition (Intact/Scrambled) or their comprehension of **verifiable story details**”

Discussion (page 15): “Critically, these lingering influences were strongest amongst participants who reported being ‘transported’ into the story world, **arguably reflective of deep processing**³⁴, regardless of the objective coherence of the text they read (Figure 5).”

Moreover, we have now rewritten the relevant section of the Results (Transporting stories linger) to additionally clarify this point in the revised manuscript (see page 11).

2. Even in the word scrambled condition, there is some evidence of self-report lingering (~2.5 out of 7). I wonder if this type of lingering may be of a different nature compared to the self-report lingering in the narrative condition. For example, I imagine if I was a participant, I may continue to wonder and think about what actually happened in the story in the word scrambled condition (out of confusion), which seems to be different from the type of lingering that are described by participants in the narrative condition. Given participants provided a description of their mental experience, I wonder if the content of the lingering could potentially be informative and distinguish across the three groups?

We thank the reviewer for bringing up this interesting point. The descriptions were quite short on average and varied in quality across participants, so a rigorous analysis of these data was not feasible. Also, the majority of participants in the Word-scrambled condition did not report any lingering. However, for those that did, we provide a synthesis of our impressions below.

Some participants described a general change in difficulty coming up with words after the reading task. For example:

- “It seemed harder after the text”
- “I think I was free flowing at first, and very confused and cluttered thoughts after.”

Others described it as easier:

- “I felt like I had more ideas quickly during the second round”
- “I felt like the words came more fluently into my head.”

Many describe feeling confused or tired and that having lasting effects:

- “confusion over what I read”
- “Maybe just tired and harder to think.”
- “I literally had a headache by the second one.”

There were also relatively few descriptions that implied content from the story persisted in mind. In reading these descriptions, they implied that the lingering consisted of random words coming to mind:

- “some words from the random sentences came to mind and seemed to interfere with my association process.”
- “I don't think there were any. I do;t think I was really influenced by the story in between even though a word or 2 from it did end up in there. I may have been kind of trying to avoid doing that too though”
- “I felt like my vocabulary changed after reading the story”

Lingering in participants who read the Intact and Sentence-scrambled conditions, instead, was described as recurrent thoughts about events from the story, persistent themes, or lasting moods. In the Word-scrambled condition, one participant explicitly stated they understood the story to some degree reported a similar experience:

- “i was confused but basically got what happened and it was sad so i had sad thoughts”

Overall, it seems as if lingering in the Word-scrambled condition may be less yoked to the situation-level meaning of the text itself as compared to lingering in the Intact or Sentence-scrambled conditions. Though, to avoid overspeculation in the manuscript itself, we have chosen not to include this discussion. Instead, we have made the descriptions themselves publicly-available for the interested reader to peruse (see SI: Supplemental Materials for a link to the OSF repository).

This examination of the experience of participants in the Word-scrambled condition also prompted us to re-examine the variability in narrative transportation. In other words, what does it mean for someone to be “transported” in the Word-scrambled condition? To this end, we plotted the average ratings for each individual item in the transportation scale for all participants from the Carver (Original) dataset in Experiment 1 (see SI: Supplemental Results III). Of note, an item indicating that the participants felt “mentally involved” in the text was a clear contributor to transportation in the Word-scrambled condition. Being “mentally involved” may be driven by general task engagement rather than immersion in a story-world and consequently helps clarify the composition of transportation for these participants. Furthermore, we conducted an additional feature selection analysis to highlight the elements of transportation that best predicted self-reported lingering in each condition of our experiments (see SI: Supplemental Results XIV). We hope that these analyses will help the interested reader get a concrete sense of the potential psychological differences between our experimental conditions.

3. Deeply processed words linger section - Is ANOVA the most optimal way of comparing across conditions? The data does not seem to be normally distributed.

We thank the reviewer for catching this. In the revised manuscript, we have switched all analyses on self-reported lingering (i.e., a 7-pt Likert scale) from ANOVAs and t-tests to their nonparametric counterparts: Kruskal-Wallis ranked sum tests and Mann-Whitney U tests, with ranked epsilon squared ($\epsilon^2_{\text{ranked}}$) and ranked biserial correlation (r_{rb}) as

estimates of effect size, respectively. The results remain consistent with the original manuscript.

4. Can authors address the magnitude of the differences observed across the different results in the discussion? For example, while transportation and comprehension scores show more notable group differences (Fig S1/2), theme similarity shows less noticeable pre-post differences even when those differences are significant. The duration of the lingering effect also seems less impactful in the real world given the maximum detectable difference lasts approximately 2.5 minutes in the study, which I understand is a result of the experimental design.

We thank the reviewer for pointing out this issue. To help explain the variability in the magnitude of our effects, we have included an additional Limitations section (see below).

Limitations (page 17-18):

“Limitations

While our data provide empirical evidence that aspects of the narrative-level meaning of a text can persist in our spontaneous thoughts, there are some important limitations. First, our free association task operates at the level of words, which is almost certainly not the level of representation of our spontaneous thoughts, nor are words likely to be what is lingering in mind after we read a story. Instead, introspection would suggest that when a story lingers in mind, the precise nature of what is persisting can be better described as a more complex kind of mental representation (e.g., events, situations, topics, themes, emotions; see *SI: Supplemental Results XII*). As a result, our objective measures of lingering (i.e., document classification and theme similarity) only indirectly reflect a subset of the actual subjective experience of lingering. Consistent with this point, the correlation between theme similarity (Post- minus Pre-story) and self-reported lingering was fairly small when calculated across all datasets in Experiment 1 ($r = 0.25$). We opted for a free word association task as the semantic associations between spontaneously generated words (i) are thought to sample the more complex mental constructs that form the basis of our thoughts⁷⁴ and (ii) are readily quantified using tools from natural language processing³². While our approach provides a reliable way to quantify lingering without self-report, future studies using more unconstrained “think-aloud” paradigms⁷⁵ and richer language embeddings grounded in agency⁷⁶ may improve the correspondence between our objective measures and subjective experience. Another limitation of this study is the lack of an objective index of processing depth. This issue been leveraged against the original levels of processing framework⁷⁷ and it limits the extent to which we can truly determine that processing depth drives lingering. Nonetheless, the sensitivity of regions of the DMN to the

situation-level meaning of narratives^{30,35,39} provides a potential way for future studies to leverage brain activity as a quantitative index of deep processing.”

Reviewer #3 (Remarks to the Author):

Bellana and colleagues present a manuscript which describes the properties that determine what types of information prospectively persist to shape cognition. Using both self-report and spontaneous word generation measures, the authors show that information lingers more persistently when it is encoded more deeply, perhaps representing greater transportation into the narrative context. situation level meaning. In brief, the authors found that intact narratives resulted in more lingering thoughts than randomly assorted words, and to a lesser extent randomly assorted sentences. In follow-up studies, the authors showed that this effect generalized to other stories, and could be manipulated by participants' goal orientation using a word list paradigm. While I think, the authors are demonstrating a very interesting phenomenon overall, there were elements of the design, analysis, and interpretation of the findings that left me “hanging” on what the underlying mechanism was. In some ways, the paper felt like it showed me a phenomenon, but fell short of telling me what cognitive operations make this phenomenon occur. To be clear, I think the authors are onto something quite awesome, but it left me wanting more (because it is so cool!). These comments and other are detailed below.

Dr. Murty – we would like to thank you for your thoughtful and enthusiastic assessment of our work. We are happy to hear that you find the phenomenon interesting. Your concern about the underlying mechanisms is crucial and we sought to address this issue with additional experiments and revised discussion.

1. In general, the overall theoretical framework seemed a little diffuse. For example, I couldn't really grasp if this was a study on extending semantic priming or understanding contextual influences on prospective memory. In some ways, I found more utility in the practical real-world examples than the underlying theory they were based on. I think not having a solid understanding of what we learn from asking this specific question, dampened the impact of the findings. Specifically, I am not sure if the authors are targeting semantic similarity, depth of processing, transportation, schemas, or situation models. However, as I write this, I get how difficult it is to operationalize these intersecting constructs.

We would like to thank Dr. Murty for highlighting this essential point. In our original manuscript, we failed to explicitly specify the relationship between the construct of 'narrative transportation' and 'deep processing', which we believe is at the heart of the confusion. Our position, now made explicit at several points in the manuscript, is that:

- (i) deep processing results in lingering, and
- (ii) the experience of being 'transported' into the world of a story entails the act of building, representing and engaging with a situation model from the text, or in other

words, *deep processing* (e.g., Busselle & Bilandzic, 2008: “Transportation into a narrative can be seen as the extent to which an audience member becomes absorbed into the activity of constructing mental models”, page 261).

A similar point was brought up by Reviewer 1 (point #1), and we ask that you refer to our earlier response for a more detailed discussion of this point.

Furthermore, we have added an additional discussion of a mechanistic explanation as to *how* deep processing might cause a recent experience to persist in our minds.

Discussion (page 16). **“Precisely *how* deep processing results in lingering remains an open question⁵⁰. If we consider all of the individual units of our knowledge and experience as nodes on a graph, deep processing may be operationalized as a learning function that results in lasting increases in the edge weights between the nodes of an input and the nodes of related knowledge and experiences. If we then model spontaneous thought as a random walk on such a network⁵⁴, we should be more likely to traverse these deeply processed edges again (i.e., lingering). Furthermore, the consequences of deep processing according to this model would not be limited to non-volitional random walks, but would also benefit more rule-based search processes on the network, consistent with the well-documented benefits of deep processing on tests of explicit memory²⁴. However, how a pure associative network model could support the lingering of more complex mental representations, such as our current concerns or goals⁵⁵ remains unclear. Agent-centered models^{56,57} that combine decisional and episodic memory processes may be necessary to capture the real-world phenomenon in which our thoughts during one task or interaction persist for minutes into the next task or interaction. In the brain, deep processing may result in lingering via a propensity to drive activity in higher-order association cortices (e.g., regions of the DMN)^{35,36}. Higher-order association cortices possess distinctively slow-drifting intrinsic dynamics, likely due to their elevated levels of local-circuit and inter-regional recurrence^{58,59}. Thus, if deep processing especially involves these brain regions, they are well-placed to generate lasting neural reverberations and, perhaps, lingering mental contexts.”**

We hope that this revised discussion, alongside our new empirical evidence that deep processing indeed results in lingering (see our response to Reviewer 2’s second point or SI: Supplemental Results IX – *Shallow processing of a coherent text reduces lingering [Experiment 4]*), helps clarify the cognitive and neural operations that underlie our results.

2. One theoretical framework that especially felt unsupported was the focus on transportation. While my gut feels like this is a good construct to study, I wanted to know why it was selected over other alternative explanations (i.e., generation of situation models; schema consistency). Further, I think the data would be more impactful if I could understand how their findings support one theoretical framework over others.

As described in our response to Dr. Murty's previous point and Reviewer 1's first point, in the revised manuscript we now explicitly define transportation as our self-report index of deep processing. The specific changes to the text are described in more detail in our response to Reviewer 1.

Our operationalization of deep processing via the transportation metric is based on theoretical models of narrative transportation, which conceptualize it as "the extent to which an audience member becomes absorbed into the activity of constructing mental models" (Busselle & Bilandzic, 2008; page 261). In other words, transportation reflects the experiential component that accompanies the act of constructing and engaging with situation models.

In the original manuscript, we describe how engaging with situation models can be thought of as an example of deep processing:

Discussion (page 16): "The notion that parsing situation-level meaning is an example of deep processing is consistent with hierarchical models of discourse comprehension. Kintsch (1998) conceptualized comprehension as a multilayered system, beginning with a 'surface code' to provide a verbatim representation of a text's words and syntax, and ending with a 'situation model' that summarizes the broader happenings they describe. The act of constructing and elaborating on a situation model requires a reader to move beyond the text itself and consider its deeper structure³⁴"

Therefore, rather than being separate from concepts such as the 'generation of situation models' or 'schema consistency', narrative transportation reflects experiential component of constructing and engaging with situation models. We take transportation as a self-reported index of deep processing. In our revisions, we have now tried to make this rationale clearer and more explicit; for specific changes, please see our response to Reviewer 1's first point.

3. I was a little concerned about individual's goal orientation during the main task in Experiment 1. Picking a cue that was embedded in the story seems like there is a bias to have individuals perform some form of memory retrieval. The reason this is problematic, is that if that arena had a "flavor" of retrieval, the findings wouldn't be very different from prior work done on semantic biases on memory retrieval. One area that could be strengthened to address this point is to find a way to quantify participants' open ended reports, and then get a metric of volitional retrieval to use as a potential moderator or confounding variable.

We strongly agree with Dr. Murty that our participants' goal orientation during free association required further consideration in the manuscript. A version of this point was also brought up by Reviewer 1 (point #3) and Reviewer 2 (point #1). As described above, we addressed this point in two ways.

First, we conducted a new preregistered experiment (Experiment 4; <https://aspredicted.org/xd38t.pdf>) where our free association task used a neutral rather than story-related cue (i.e., "Enter a word to begin!"). As outlined in our response to Reviewer 1 (point #3), we were able to replicate our findings in terms of self-reported lingering, document classification and theme similarity in the absence of a story-related cue (see SI: Supplemental Results VIII – *Lingering is not a result of using story-related cues [Experiment 3]*). These data provide strong evidence that our results are not an artifact of using a story-related cue.

Second, we probed the volitional nature of lingering by additionally querying participants in our new experiments (Experiment 3: Neutral cue and Experiment 4: Emotion vs. Proofread; n = 240). As outlined in our response to Reviewer 2 (point #1), the majority of participants described lingering as unintentional rather than intentional (see SI: Supplemental Results XI – *Is lingering volitional? [Experiments 3 & 4]*). This suggests that lingering is not an active rehearsal process nor does it tap directly into the kinds of explicit memory processes that typically benefit from deep processing. Unfortunately, due to the varied length and quality of the open-ended descriptions from our Experiment 1 and 2, we were unable to reliably determine the volitional nature of lingering in the data of our previous participants. We have included the relevant figure below, for your convenience.

Figure S15. *Lingering may be outside of our volitional control.* Participants in Experiment 3 and 4 answered additional questions about the volitional quality of their experience of a story lingering in mind, if they experienced any lingering at all. Most participants endorsed lingering as unintentional, distinguishing it from rehearsal and the kinds of explicit memory that traditionally benefit from deep processing (Craik & Lockhart, 1972). Plots were generated in R using `waffle()` (Rudis, 2017).

Together, we believe these results provide strong evidence that lingering as reported in this manuscript can be distinguished from more explicit memory retrieval processes.

4. I was a little stuck on how to interpret the findings from the classifier. In study 1, this classifier didn't disambiguate from sentence scrambles and intact story. Then, in the follow-up, the classifier was efficacious for some but not all stories. Then in Experiment 2, which I found the most surprising, was that the classifier was unable to disambiguate between conditions. The authors then followed up with a theme similarity analysis which showed a difference for the story condition, but also the theme condition. This pattern of results made it hard for me to interpret what unified interpretation was driving the lingering effects. I also wanted more consideration on what factors may lead to explicit versus implicit lingering, and how this differed across studies.

We agree with Dr. Murty that the heterogeneity in our results, particularly across our two objective measures of lingering, warrants further discussion. To this end, we included a paragraph in the Methods section describing an important distinction between document classification and theme similarity:

Methods: Analysis of free association chains (page 21):

“Note that our two approaches for analyzing free association data, theme similarity and document classification, may not always coincide. While theme similarity makes use of a continuous estimate of semantic similarity between any two words available to the pretrained model³², our implementation of document classification relies on exact word matches across participants. Consider the following example: all participants in the document classifier’s training set have the word “stream” in their post-story chains, while none have it in their pre-story chains. Also, the semantically-related “creek” was not used by any participants in the training set during pre- or post-story. Now, if a held-out participant uses the word “creek” post-story, but not “stream”, the classifier will not be able to make use of the importance of the word “stream” to predict that “creek” should also imply the chain was generated post-story. However, in the context of theme similarity, instances of the words “stream” and “creek” will both show strong semantic similarity to the story theme “river”, and thus “creek” will contribute signal to our theme similarity analysis but not to document classification.”

Therefore, in Experiment 2, participants in the Story and Theme conditions may have used words that were related to the story themes after performing the task, but these words may have been more heterogeneous across participants. Considering the task required participants to try and extract meaning from an ambiguous input, it seems reasonable that the semantic/situational content being accessed in the Word-list task was more heterogeneous across participants as compared to an Intact story in Experiment 1.

In addition, we have now included a Limitations section in the revised manuscript that discusses the apparent differences between our strong effects of self-reported lingering and relatively weaker indices of objective lingering (see page 16 in the Manuscript). In brief, our free association task concerns word generation, while the phenomenon of lingering itself is unlikely to operate at the word-level. Therefore while reliable, our objective measures of lingering may not properly capture the full, rich experience of lingering.

In line with the above point, our additional querying of participants’ experience of lingering in Experiments 3 and 4 suggested that, beyond changes in topics, participants’ post-story free association was also characterized by (i) the ease/difficulty with which words came to mind and (ii) changes in the emotions that they felt (see below for an excerpt from the revised supplement; SI: Supplemental Results XII – *What does lingering feel like? [Experiments 3 & 4]*).

Figure S16. *Content of lingering (Experiment 3).* Participants in Experiment 3 answered additional questions about what constituted lingering in their experience. Participants tended to describe lingering as a change in the topics that came to mind during post-story free association, but also a change in the ease/difficulty of coming up with words and/or a change in the emotions they felt. Participants were less likely to describe it as boredom or tiredness.

These phenomenological changes may not have always been reflected in the content of free (word) association. Therefore:

Limitations (page 16): **“While our approach provides a reliable way to quantify lingering without self-report, future studies using more unconstrained “think-aloud” paradigms⁷⁵ and richer language embeddings grounded in agency⁷⁶ may improve the correspondence between our objective measures and subjective experience.”**

5. Did the authors collect data on transportation in the word list study. This seemed like a really promising construct, and I am curious how it would differ across the story and them conditions in Experiment 2.

We thank Dr. Murty for bringing up this point. Yes – in the Supplemental Information (now, SI: Supplemental Results VII – *Effects of deep processing on comprehension and transportation [Experiment 2]*) of the original manuscript, we showed that participants in the conditions associated with deeper processing (i.e., Story and Theme vs. Tangible and Italic) reported feeling more transported into the story/task.

Figure S9. *Deeper processing leads to more transportation.* Distribution of transportation scores for all participants across conditions in Experiment 2. Conditions where participants encoded the list of the words more deeply (i.e., Story and Theme) were associated with more transportation into the world of the story as compared to shallow processing conditions (i.e., Italic and Tangible). Transportation was assessed using a modified version of the Narrative Transportation Scale (Green & Brock, 2000) and a proportion of the maximum attainable score (max = $7 \times 12 = 84$). Given that participants did not read a story, transportation was measured after informing participants that the word list they read contained a “hidden story”, and each transportation question referenced their subjective experience of this hidden story. Each point represents a participant. Black points represent condition means. Error bars reflect 95% confidence intervals. Dashed horizontal line represents the lowest attainable score (each question received a rating of 1 to 7; lowest attainable score = $12/84 = 0.14$).

While transportation in the Theme condition was numerically lower than that of Story, this difference was not statistically significant [$t(158) = -1.46, p = 0.146, d = -0.23$]. This point, in addition to the comparable performance in free recall (see SI: Supplemental Results VI – *Validating the levels of processing manipulation with free recall [Experiment 2]*) suggests that depth of processing may have been similar across these two conditions. In line with this account, participants in these two conditions reported comparable levels of self-reported lingering and theme similarity in free association (Manuscript, Figure 6).

The paragraph immediately above (“While transportation in the Theme...”) was included in the revised Supplemental Information section to better emphasize the theoretical implications of these results.

6. The authors state that there is little empirical evidence to support our intuitive sense that 'meaningful' experiences resonate with us, but this precludes a fairly large literature on intrusive memories that could be incorporated into the manuscript.

We thank Dr. Murty for pointing out our oversight. We have included citations of papers from the intrusive memory literature in our revised manuscript. Unfortunately, due to limitations in the number of citations (we are already over the suggested 70 citations), we were limited in our ability to provide an in-depth treatment of this literature.

Introduction (page 3): "More generally, social information^{17,18} and emotional experiences¹⁹⁻²¹ appear to exert long-lasting influences on our mental context, in many cases intruding on our thoughts against our will^{22,23}."

7. The authors discuss the idea of levels of processing as an organizational principle, which I thought sounded like a promising framework. Given this, additional experiments that tested predictions made from the lens of levels of processing (i.e., transfer appropriate processing) could help strengthen the findings.

We thank Dr. Murty for bringing up this interesting possibility. We think transfer appropriate processing is a compelling new direction to take this line of work, one that we have been thinking about ourselves for a little while now. An intriguing idea may be that the default 'level of representation' at which our spontaneous thoughts operate may be one at the level of situations. Therefore, engaging with situation-level meaning (e.g., the plot of a narrative) will have a propensity to carry-over into the content of our spontaneous thoughts, consistent with an account of transfer appropriate processing. However, we believe that a proper examination of transfer appropriate processing is its own endeavor and would be outside the scope of the present manuscript.

Nonetheless, we conducted an additional experiment examining a different prediction associated with the levels of processing framework: *shallow processing should reduce lingering, even when reading coherent stories*. To this end, we had participants read an intact version of the Carver story while performing one of two cover tasks: (i) judging the valence of each sentence in the story (Emotion condition; deep processing) and (ii) counting the number of typos and/or font errors in each sentence (Proofread condition; shallow processing). In line with our levels of processing-based account, the Proofread condition was indeed associated with reduced lingering. The results from Experiment 4 are outlined in more detail in our response to Reviewer 2 (point #2) as well as in the Supplemental Information for the revised manuscript.

8. Did the authors consider doing some sort of cross story, theme analysis? If themes are the major focus of the lingering effect, perhaps a classifier could be fit to disambiguate free associations drawn from different stories, with the prediction that there would be more errors in the two re-writes of the same story versus across story comparisons.

We thank Dr. Murty for highlighting this potential use of cross-story classification. Rather than training a classifier to predict which story participants had read based on their free association (which we believe is what was suggested), we tested the hypothesis that lingering is specific to a story's themes using related analysis that parallels the structure of our other classifiers. Specifically, if lingering is story-specific, a classifier trained to discriminate between pre- and post- story free association chains in participants who read the original Carver story should:

- (i) perform *above chance* at classifying pre- vs. post-chains in participants who read a different version of the same story (e.g., Carver-Rewrite), but
- (ii) *fail to achieve above chance classification* in participants who read a different story (e.g., July)

We report the results of this classifier in the supplement (SI: Supplemental Results XV – *Cross-story classification as evidence that lingering is story-specific [Experiments 1 & 3]*). In line with our hypothesis, a classifier trained on free association data from the Intact version of Carver-Original was able to discriminate between pre- and post-story chains above chance for participants from the Carver-Rewrite dataset (58% classification accuracy, permutation test: $p = 0.002$) but not the July dataset (53% accuracy, $p = 0.17$) (Figure S19, see below). For reference, the Carver-Original classifier in Experiment 3 was able to classify pre- vs. post-story chains with 77% accuracy ($p < 0.002$) when applied to participants from the same dataset. Reduced classification is expected given that participants in Carver-Rewrite a substantially edited version of the story (see Supplemental Methods) and participants who read Carver-Rewrite also performed free association using story-related cue words (“body” and “water”) as compared to the neutral cue from Experiment 3 (“Enter a word to begin!”). In spite of these difference, our classifier only exceeded chance performance when it was tested on participants who read a version of the same story, consistent with the claim that lingering is associated with the persistence of story-specific content.

Figure S19. *Cross-story classification as evidence that lingering is story-specific.* Histograms of how accurately a document classifier could discriminate between pre- and post-story free association chains across separate datasets. Classifiers were trained on data from participants who read the Intact version of Carver-Original in Experiment 3 (Neutral cue variant; $n=80$). Classifiers were then tested on a separate group of participants who read a different version of the same story (i.e., Intact version of Carver-Rewrite; $n = 80$) or a different story entirely (i.e., Intact version of July; $n = 80$). Dashed lines represent the mean classification accuracy. Null distributions were estimated by randomly shuffling the labels of word chains (pre, post) in the test set and recalculating classification accuracy over 500 permutations. Likelihood of achieving mean classification from the null distribution was calculated using a permutation test [*ns* $p > .05$; * $p < .05$; ** $p < .01$; *Note:* minimum p-value estimate for this analysis is $p < 0.002$].

REVIEWERS' COMMENTS

Reviewer #1 (Remarks to the Author):

The authors have done a careful and thorough job of addressing my (and the other reviewers') concerns. I have no further concerns, and I recommend this manuscript for publication in its current form. I think it will make a strong contribution to the literature.

Reviewer #2 (Remarks to the Author):

Thank you to the authors for their elaborate effort in addressing the concerns raised. The additional experiments implemented provided converging results overall, which further strengthen their conclusion. The more in-depth description of the concepts in the introduction in response to my comments as well as other reviewers' also better contextualize their study and highlight its novelty. I also appreciate the newly added limitations section and the level of details included in the Supplementary. The paper has improved considerably and I have no remaining comments.

Reviewer #3 (Remarks to the Author):

I was truly delighted to see the authors thoughtful responses and revisions to this manuscript. I found the introduction to be extremely compelling, and the connections of levels of processing with transportation to be invigorating, and thoughtfully tested. I also really appreciated the new experiments, and the additional analysis, which alleviate my concerns. I am really excited about this work!